# CMOS backend-of-line compatible memory array and logic circuitries enabled by high performance atomic layer deposited ZnO thin-film transistor

Wenhui Wang[1,5], Ke Li[1,5], Jun Lan [1], Mei Shen[1], Zhongrui Wang [2], Xuewei Feng[3], Hongyu Yu[1], Kai Chen[1], Jiamin Li[1], Feichi Zhou[1], Longyang Lin [1]✉, Panpan Zhang [4]✉ & Yida Li [1]✉

The development of high-performance oxide-based transistors is critical to enable very large-scale integration (VLSI) of monolithic 3-D integrated circuit (IC) in complementary metal oxide semiconductor (CMOS) backend-of-line (BEOL). Atomic layer deposition (ALD) deposited ZnO is an attractive candidate due to its excellent electrical properties, low processing temperature below copper interconnect thermal budget, and conformal sidewall deposition for novel 3D architecture. An optimized ALD deposited ZnO thin-film transistor achieving a record field-effect and intrinsic mobility ($\mu_{FE}$ /$\mu_o$) of 85/ 140 cm$^2$/V·s is presented here. The ZnO TFT was integrated with HfO$_2$ RRAM in a 1 kbit (32 × 32) 1T1R array, demonstrating functionalities in RRAM switching. In order to co-design for future technology requiring high performance BEOL circuitries implementation, a spice-compatible model of the ZnO TFTs was developed. We then present designs of various ZnO TFT-based inverters, and 5-stage ring oscillators through simulations and experiments with working frequency exceeding 10's of MHz.

With the growing demand for data-driven applications such as the next-generation machine learning accelerators and the Internet of Things (IoT), the traditional von-Neumann architecture with disjoint memory and processing units suffers from huge memory latency and limited data bandwidth, which is further intensified by the scaling limits of silicon transistors. In order to surpass these bottlenecks, monolithic-three-dimensional (M3D) integration of fused logic and memory[1], or in-memory computing[2,3], with functional logic circuits has emerged as a potential solution. However, the utility of silicon (Si) technology for beyond BEOL integration is challenged by the low thermal budget (<400 °C) posed by the low-k dielectric and copper

interconnects, where subjected to higher temperatures results in reliability issues. Beyond-Si devices that can be co-integrated additively on Si-based complementary metal–oxide–semiconductor (CMOS) chips, include carbon nanotube (CNT) field-effect transistors (FETs)[4–6], two-dimensional (2D) materials[7–10], and oxide semiconductors[11,12]. Among them, oxide semiconductors, such as indium gallium zinc oxide (IGZO)[13–15], indium oxide (In$_2$O$_3$)[16], and zinc oxide (ZnO)[17–22], are well poised to be competitive n-channel materials beyond silicon due to their low thermal budget process, good transparency, process maturity for large scale deposition, and decent electrical properties, such as high carrier mobility, wide bandgap, low

[1]School of Microelectronics, Southern University of Science and Technology, 518055 Shenzhen, China. [2]Department of Electrical and Electronic Engineering, The University of Hong Kong, 999077 Hong Kong SAR, China. [3]Shanghai Jiao Tong University, 200240 Shanghai, China. [4]State Key Laboratory of Information Photonics and Optical Communications, Beijing University of Posts and Telecommunications, 100876 Beijing, China. [5]These authors contributed equally: Wenhui Wang, Ke Li. ✉e-mail: linly@sustech.edu.cn; tanji_ic@bupt.edu.cn; liyd3@sustech.edu.cn

gate leakage[23,24]. These merits make them suitable for backend-of-line (BEOL) integration as memory drivers in memory-centric computing cells or high-performance thin-film transistor (TFT)-based BEOL logic circuitries (Fig. 1a)[7,25]. With the increasing interest to realize new computing architecture with new functionalities and enhanced computing power, the development of high-performance oxide-based transistors to enable very large-scale integration (VLSI) of M3D integrated circuit (IC) in CMOS BEOL is timely.

Of these promising n-channel oxide candidates, polycrystalline ZnO exhibits one of the highest carrier mobility through proper crystallinity and oxygen vacancies ($V_O$) tuning[17,22]. In addition, its other advantageous properties, such as wide direct bandgap (~3.3 eV) and high thermal conductivity, make ZnO a strong competitor to complement the matured and conventional silicon. Moreover, ZnO has been reported to be grown by industry-accepted techniques such as magnetron sputtering[26], or atomic layer deposition (ALD)[17]. All these benefits offer ZnO as one of the best propositions in low-temperature channel material selection for CMOS-BEOL integration. While there have been reports on high-performance ZnO TFT[27], the reproducibility

of results is still challenging due to the difficult process control as well as ensuring the long-term stability of the ZnO layer due to its hygroscopic nature. Further, among the popular deposition techniques, the ALD approach is attractive for its low-temperature process (typically <300 °C), accurate stoichiometry, thickness and uniformity control, and conformal sidewall deposition for M3D integration of vertically integrated architectures[28,29]. However, despite the promises, reports on high-performance ALD ZnO TFT, and a systematic guidance to implement it in BEOL-compatible logic circuit are still lacking, calling the need to further address these aspects.

High-performance oxide TFTs have been reported to be good selectors for emerging memories such as resistive random access memory (RRAM)[7], and can be used to implement circuits such as basic logic gates[30], amplifiers[31], gate driver[32], microprocessor[33], etc. However, from logic circuit point of view, unipolar oxide TFTs-based circuit designs face huge challenges due to the lack of p-type oxide semiconductor FETs that match that of its n-type counterpart[34]. In order to overcome the challenges of designing for unipolar device-based circuits, pseudo-CMOS design style, including pseudo enhancement and

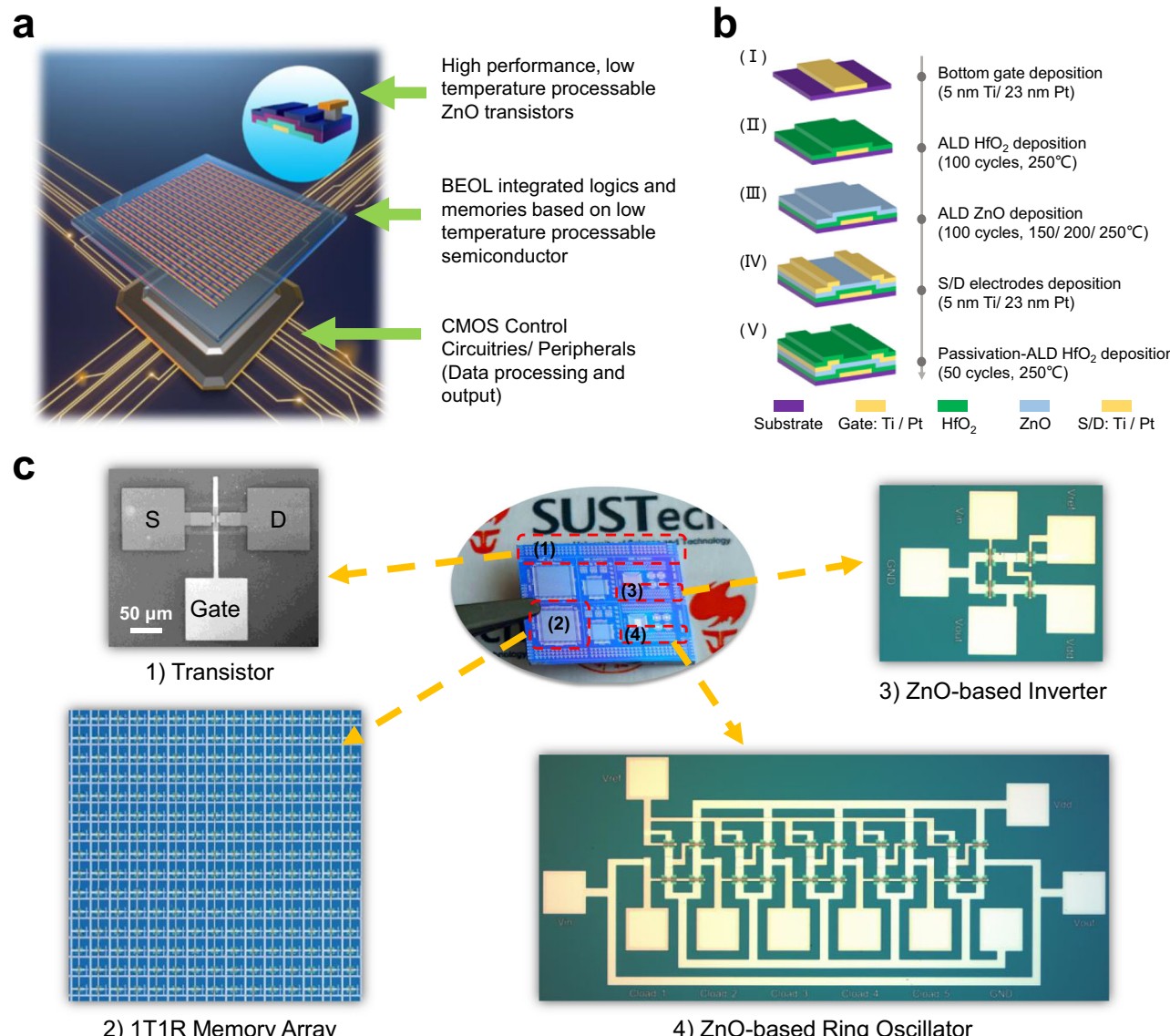

**Fig. 1 | ZnO as CMOS-BEOL-compatible transistor material. a** Schematic illustrating the use of low-temperature processable ZnO semiconductor, allowing for BEOL integration of logic circuits and memory array. **b** Schematic illustrating the fabrication process flow of the ZnO TFT. **c** Photo image of the fabricated sample containing the ZnO TFTs, 1T1R memory array, inverter, and ring oscillator.

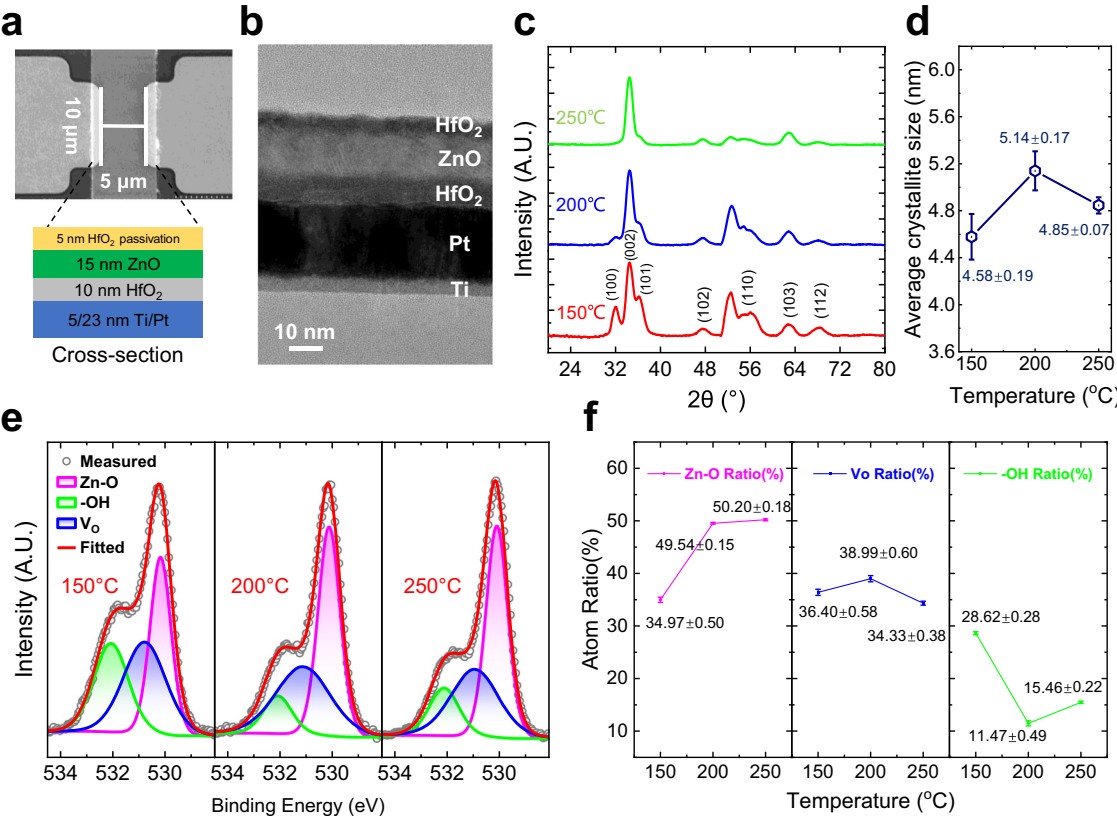

**Fig. 2 | Material characterizations of ZnO channel layer. a** SEM image of the ZnO TFT channel and cross-section schematic. **b** TEM image of the ZnO TFT stack, with the different layers labeled. **c**, **d** GI-XRD scan, and average crystallite sizes (with error bar) calculated from the Scherrer's formula, respectively, of the three different temperatures deposited ZnO films. **e**, **f** O1s XPS spectra of the ZnO channel layers deposited at three temperatures as indicated, and plot showing the Zn−O, $V_O$, and −OH atomic percentages (with error bar) as a function of the three different temperatures used[12,17,49–51].

pseudo depletion design, is a popular design method, which applies two-stage structures to achieve full $V_{DD}$ swing at the output voltage ($V_{OUT}$)[25,35–37]. However, these efforts are currently lacking, and systematic investigation is necessary for future implementation.

In this work, we report on a stable, high-performance ZnO TFT that exhibits one of the highest reported field-effect mobility ($\mu_{FE}$) of 85 cm²/V·s using ALD process with performance stability exceeding 90 days in an unpackaged form. This was achieved with an optimized ALD deposition temperature (200 °C), and with a 5-nm-thick HfO₂ passivation layer. Excellent electrical properties, including low positive threshold voltage ($V_{TH}$, 0.72 V), negligible hysteresis (<50 mV), and low $D_{it}$ ($2.45 \times 10^{11}$ eV⁻¹ cm⁻²), were also achieved. Through careful X-ray photoelectron spectroscopy (XPS) and X-ray diffraction (XRD) characterizations, we correlate the electrical properties of the ZnO TFT with the amount of $V_O$ concentration and crystallites size, elucidating such effects on the performance for possible future large-scale implementation. To demonstrate the ZnO TFT as a capable memory driver, we co-integrated it with HfO₂ RRAM into a functional 32 × 32 1T1R array; this shows its suitability for BEOL integration in analog computing. In order to co-design for unipolar TFT-based circuits, we first presented on the compact modeling of our fabricated ZnO TFT, followed by the Pseudo-CMOS design methodology to capture and accommodate the material and device limitations; these were all achieved by cross-validation of simulation and experimental results. Process variation-aware simulation framework was then implemented to evaluate the performance and robustness of ZnO TFTs-based circuits. Experimentally, the pseudo enhancement load inverter (PEL), linear enhancement load inverter (LEL) and conventional depletion load inverter (DL) based on unipolar ZnO TFTs were fabricated and characterized[35,38]. To further explore the merits of high-mobility ZnO

TFTs at the circuit level, the 5-stage ring oscillators (ROs) based on these three different types of inverters were designed and demonstrated. The results from this work are expected to provide guidance for future implementation of ZnO TFT-based circuitries at BEOL.

## Results

### Fabrication of TFT and circuits

Staggered, bottom-gate ZnO TFTs were investigated in this work. The bottom-gate electrode composed of Ti/Pt (the thickness is 5/23 nm) was first deposited by e-beam evaporation (EBE) onto a 285 nm SiO₂ layer on Si substrate. Following, 10 nm thick HfO₂ layer was deposited as gate dielectric by ALD with H₂O as the oxygen source at 250 °C. 15 nm thick ZnO active channel layer was then deposited by ALD at 3 different temperatures −150/200/250 °C. The effect of the deposition temperatures on the TFT performance is discussed later. Both the gate dielectric and ZnO channel regions were defined via standard lithography followed by a buffered oxide etch (BOE). Then, the source/drain electrodes (Ti/Pt − 5/23 nm) were deposited using EBE followed by a lift-off process. After that, a 5 nm thick HfO₂ layer was deposited by ALD to passivate the channel region. Finally, the HfO₂ layer on the contact pads was removed using standard lithography, followed by a BOE etch for electrical measurements. In this work, ZnO TFTs with channel widths ($W_{CH}$) of 10 μm, and channel lengths ($L_{CH}$) of 2/5/10 μm were fabricated. The detailed fabrication process flow as described is illustrated in Fig. 1b. Details of the process parameters are provided in "Methods".

To demonstrate the use of ZnO TFT as a RRAM selector in a 1T1R array, a 1 kbit (32 × 32) 1T1R array was designed. An additional 5 nm thick HfO₂ was deposited as the switching layer by ALD at 250 °C after source/drain electrode deposition of TFT, followed by Ti/Pt top

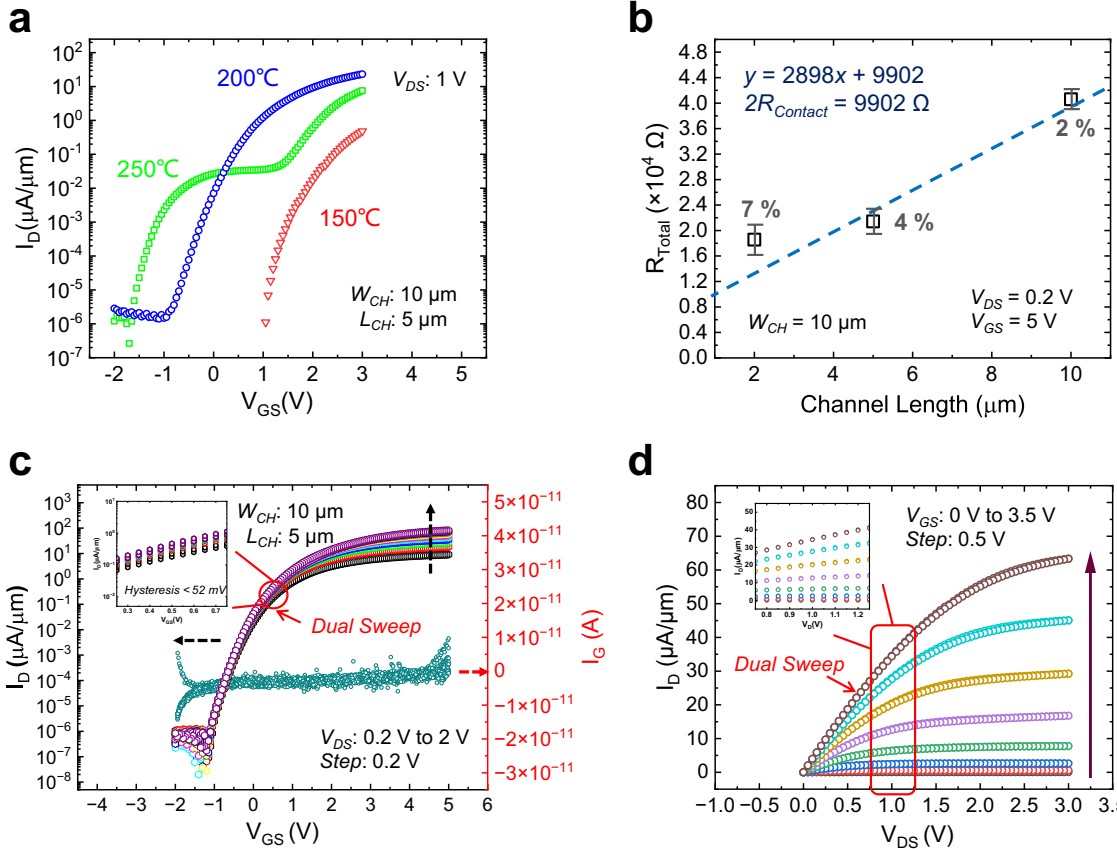

**Fig. 3 | ZnO TFT electrical performance. a** Transfer curves ($I_D$-$V_{GS}$) of the ZnO TFTs fabricated at the three different temperatures with $W_{CH}$ and $L_{CH}$ of 10 μm and 5 μm, respectively. **b** TLM measurements of the TFT with three different $L_{CH}$ (2, 5, 10 μm), with a maximum standard deviation of 7%. The $R_{Contact}$ is extracted via the vertical intercept as shown in the linear line fit equation; the $2R_{contact-specific}$ when normalized to the $W_{CH}$ of 10 μm is then obtained as 0.99 kΩ μm. **c** $I_D$-$V_{GS}$ family of curves (dual sweep) measured over different $V_{DS}$ (0.2– 2 V). A high current on-off ratio up to ~$10^8$ is obtained with $SS$ of 110 mV/dec and hysteresis <52 mV. **d** Output ($I_D$-$V_{DS}$) family of curves (dual sweep) measured over different $V_{GS}$ (0–3.5 V) of the same TFT. The zoomed-in region of the plots in (**c**, **d**) (circled) are shown in the inset, respectively, indicating the small hysteresis measured of our fabricated TFT.

electrode deposition. The size of the RRAM was 5 μm × 5 μm. In the array, ZnO TFT with the $W_{CH}$ and $L_{CH}$ of 10 μm and 5 μm, respectively, was used as the select transistor. The microscopic image of the 1T1R array together with a zoom-in image of one single 1T1R cell and the process flow is shown in Supplementary Fig. S1. To demonstrate the functionality of the ZnO TFTs-based inverters and ring oscillators, we demonstrate the designs of three different kinds of inverters, namely, PEL, LEL, and DL inverters and their respective ring oscillators. In the fabrication, the process flow was basically the same as the ZnO TFT, with only the additional step of via hole opening before top electrode deposition for overlapping bottom and top interconnects. Figure 1c shows the photo image of the fabricated sample containing the ZnO TFTs, logic circuits, and 1T1R memory array. Details of measurements and characterizations of the devices and circuits are provided in "Methods".

**Materials characterization**

Detailed material characterizations for ZnO TFT have been performed to analyze and optimize the TFT performance. Figure 2a shows the ZnO TFT-stacked structure and scanning electron microscope (SEM) image of the channel region with width (10 μm) and length (5 μm) as labeled. The cross-sectional transmission electron microscope (TEM) image shown in Fig. 2b confirms the layers thicknesses and good interface quality between the different layers. The energy dispersive spectroscopy (EDS) mapping of Ti, Pt, Hf, and Zn elements taken from the channel material stack shows clear distinction of the different

layers (Supplementary Fig. S2). Atomic force microscope (AFM) images of the $HfO_2$ dielectric and ZnO channel are shown in Supplementary Fig. S3. Small root-mean-square roughness ($R_q$) of the $HfO_2$ gate insulator (0.289 nm) and the ZnO channel (0.342 nm) are obtained, indicating the high quality of all the deposited layers, beneficial for reduced carrier scattering.

To analyze the crystallinity and the preferred orientation, grazing incidence X-ray diffraction (GI-XRD) scans were obtained for the ZnO thin films deposited at the three different temperatures, as shown in Fig. 2c. The XRD patterns show that all the ZnO films are polycrystalline with a hexagonal wurtzite structure, with the peaks identified as (100), (002), (101), (102), (110), (103), (112) phases[39,40]. The XRD patterns of all three films exhibit the enhanced intensities for the peaks corresponding to (002) plane, indicating preferential orientation along the *c* axis[40,41]. The 150 °C deposited ZnO film shows the preferable (002) orientation with additional (100) and (101) peaks. However, the (100) and (101) peaks weaken significantly, and the (002) peak dominates as the deposition temperature of ZnO films increases. This can be attributed to the low surface energy of (002) plane, which is the most thermodynamically favorable[40,42,43]. When the deposition temperature of film is high, Zn and O atoms can obtain sufficient energy to transfer themselves into energetically favorable positions, which results in ZnO thin films transiting to the (002) preferred orientation at high deposition temperature[44–46]. Figure 2d shows the average crystallite sizes of the three different ZnO thin films, which were calculated using Scherrer's formula. The crystallite size calculations are conducted

based on the different diffraction peaks measured, and by assuming that the crystallite size is the same in all directions (see Supplementary Note S1 for details). The largest average crystallite size of 5.14 nm is obtained of the 200 °C deposited ZnO film, which is favorable for high mobility due to reduced grain-boundary scattering[47].

The details of the oxygen defects in the ZnO thin films are elucidated from the XPS O1$s$ spectra of all the ZnO films, shown in Fig. 2e. The O1$s$ peaks were typically deconvoluted into three energy level sub-peaks, including a lower binding energy peak centered at ~530.1 eV (magenta) corresponding to the lattice oxygen or metal oxide (Zn–O), a higher binding energy peak centered at ~532.1 eV (green) corresponding to the loosely bound oxygen like hydroxyl groups (−OH), and the medium binding energy peak centered at ~530.95 eV (blue) attributed to non-lattice oxygen and associated with $V_O$[12,17,48–50]. The extracted at% of the Zn–O, $V_O$, and −OH bonds are shown in Fig. 2f. It can be seen that the 200 °C deposited ZnO film contains the largest amount of $V_O$, which helps to improve the carrier density for better drive current[17,49]. At the same time, it also contains the lowest impurity concentration (−OH), leading to superior electrical performance, as presented in the following section[50].

## ZnO TFT electrical characterization

Figure 3a shows the transfer curves ($I_D$-$V_{GS}$) of the ZnO TFTs fabricated at the three different temperatures with $W_{CH}$ and $L_{CH}$ of 10 μm and 5 μm, respectively. The current–voltage ($I$–$V$) curves in Fig. 3 are all normalized to the $W_{CH}$. The 150 °C deposited ZnO TFT exhibits the lowest drive current, lowest μ$_{FE}$ (3.63 cm$^2$/V·s), and largest $V_{TH}$ (2.5 V) as compared to the TFTs fabricated at higher temperatures. This suggests the detrimental effect of higher trap states, which allows unstable stray charges entering or leaving the traps, resulting in reduced μ$_{FE}$ and overly positive shift in the $V_{TH}$[40,45,51]. On the other hand, the ZnO TFTs deposited at 220 °C shows the emergence of a small hump in the subthreshold region of the transfer curve (Supplementary Fig. S4), while a distinct hump phenomenon appears in the 250 °C deposited ZnO TFT that is accompanied by a negative shift of $V_{TH}$, observable in all devices fabricated at this temperature. This phenomenon has been examined in several works, and can be attributed to the existence of dual conduction channels, i.e., a main channel and a parasitic channel that turn on a different voltage as a result of unoptimized growth

conditions[52–55]. The 200 °C deposited ZnO TFT shows the best overall performance with the highest drive current and a low positive $V_{TH}$, and is further discussed. Figure 3c shows the normalized transfer curves ($I_D$-$V_{GS}$) for $V_{DS}$ from 0.2 to 2 V ($V_{DS}$ step size of 0.2 V), with corresponding gate leakage current ($I_G$) less than $10^{-12}$ A/μm (limit of measurement setup), which indicating good quality of HfO$_2$ gate insulator. Figure 3d shows the normalized output curves ($I_D$-$V_{DS}$) at $V_{GS}$ from 0 to 3.5 V ($V_{GS}$ step size of 0.5 V). The dual sweep $I$–$V$ characteristics for both transfer and output curves show negligible hysteresis of 47 mV at $V_{GS}$ = 0.5 V, as shown in the insets of Fig. 3c, d. The hysteresis is extracted as follows: At a selected $V_{GS}$ in the transfer curve, the voltage difference between the forward and backward sweep was obtained. The extracted hysteresis as a function of $V_{GS}$ at $V_{DS}$ of 1 V as shown in Supplementary Fig. S5, with the definition of the hysteresis extraction shown graphically in the inset. The hysteresis values vary with different $V_{GS}$, and a maximum hysteresis value of 52 mV is obtained near the saturation voltage of 4.2 V. The small hysteresis value of our TFT further suggests the good interface quality with low interface traps. A high drive current of 64 μA/μm, with a large $I_{ON/OFF}$ ratio >$10^8$ was obtained. Furthermore, the fabricated ZnO TFTs show good uniformity, as can be seen from the transfer curves of 20 devices randomly selected across entire sample with minor standard deviations, as shown in Supplementary Fig. S6.

The contact resistance ($R_{Contact}$) is an unwanted parasitic in transistors, where it degrades drive current required for high-speed operation of IC chips. This is especially so at advanced technology node where the $R_{Contact}$ can form a significant portion of the entire TFT's resistance. Here, we extracted the $R_{Contact}$ of the ZnO TFT via the transfer length method (TLM), by utilizing the different channel length ($L_{CH}$) TFTs measured and plotting the TFTs' resistance vs. $L_{CH}$ (three different channel lengths are utilized here), as shown in Fig. 3b. A total $R_{Contact}$ of 9.902 kΩ was extracted at the fitting line's vertical intercept point, translating to a specific contact resistance $R_{Contact-specific}$ of 0.99 kΩ·μm when normalized to the TFTs' $W_{CH}$ of 10 μm. Further optimization of the $R_{Contact}$ value can be achieved though source/drain contact engineering, such Ar plasma treatment for effect Schottky barrier height tuning[56].

Here, we present on the extraction of both μ$_o$ and μ$_{FE}$ of the fabricated ZnO TFTs. μ$_o$ is defined as the true mobility of the ZnO TFT in

## Table 1 | Benchmark of reported ALD ZnO TFT performance parameters

| References | Year | Temp. (°C) | ZnO thickness (nm) | Oxide/thickness (nm) | I$_{ON}$/I$_{OFF}$ ratio | SS (mV/dec) | V$_{TH}$ (V) | μ$_{FE}$ (cm$^2$/V·s) |
|---|---|---|---|---|---|---|---|---|
| 45 | 2019 | 200 | 33 | SiO$_2$/30 | 2.8×10$^9$ | 127 | 4.00 | 7.8 |
| 60 | 2019 | 200 | 15 | ZrO$_2$/5 | ~10$^7$ | 69 | 0.10 | 36.8 |
| 17 | 2019 | 100 | 16 | Al$_2$O$_3$/36 | ~10$^7$ | 320 | 1.23 | 38.4 |
| 46 | 2020 | 300 | 36 | HfO$_2$/30 | 1.9×10$^7$ | 175 | 1.10 | 11.8 |
| 61 | 2020 | 200 | 33 | Al$_2$O$_3$/60 | 4.1×10$^9$ | 131 | 3.24 | 19.6 |
| 62 | 2020 | 300 | 30 | Al$_2$O$_3$/40 | ~10$^7$ | 220 | 6.00 | 16.2 |
| 49 | 2020 | 150 | 14 | Al$_2$O$_3$/40 | ~10$^8$ | 210 | 0.14 | 31.1 |
| 18 | 2021 | 350 | 11 | SiO$_2$/90 | 5.0×10$^9$ | / | 18.70 | 43.2 |
| 63 | 2021 | 100 | 25 | Al$_2$O$_3$/30 | 3.0×10$^7$ | 210 | / | 14.3 |
| 64 | 2021 | 100 | 16 | Al$_2$O$_3$/36 | ~10$^8$ | 170 | 1.05 | 31.2 |
| 50 | 2022 | 100 | 23 | Al$_2$O$_3$/36 | ~10$^8$ | 170 | 1.00 | 32.1 |
| 19 | 2022 | 100 | 15 | HfO$_2$/30 + Al$_2$O$_3$/10 | ~10$^7$ | 110 | −3.00 | 55.5 |
| 65 | 2022 | 400 | 20 | Al$_2$O$_3$/30 | ~10$^{10}$ | 225 | 3.23 | 17.9 |
| 40 | 2022 | 150 | 20 | Al$_2$O$_3$/50 | 2.0×10$^7$ | 250 | 1.41 | 10.7 |
| 20 | 2022 | 100 | 20 | Al$_2$O$_3$/40 | 4.3×10$^9$ | 243 | 1.13 | 45.3 |
| 21 | 2022 | 300 | 3 | Al$_2$O$_3$/5 | ~10$^8$ | 94 | / | 84.0 |
| 66 | 2023 | 200 | 17 | Al$_2$O$_3$/15 | ~10$^{12}$ | 75 | / | 32.8 |
| 22 | 2023 | 200 | 20 | Al$_2$O$_3$/20 | ~10$^9$ | 179 | 2.73 | 43.8 |
| This work | 2023 | 200 | 15 | HfO$_2$/10 | ~10$^8$ | 110 | 0.72 | 85.0 |

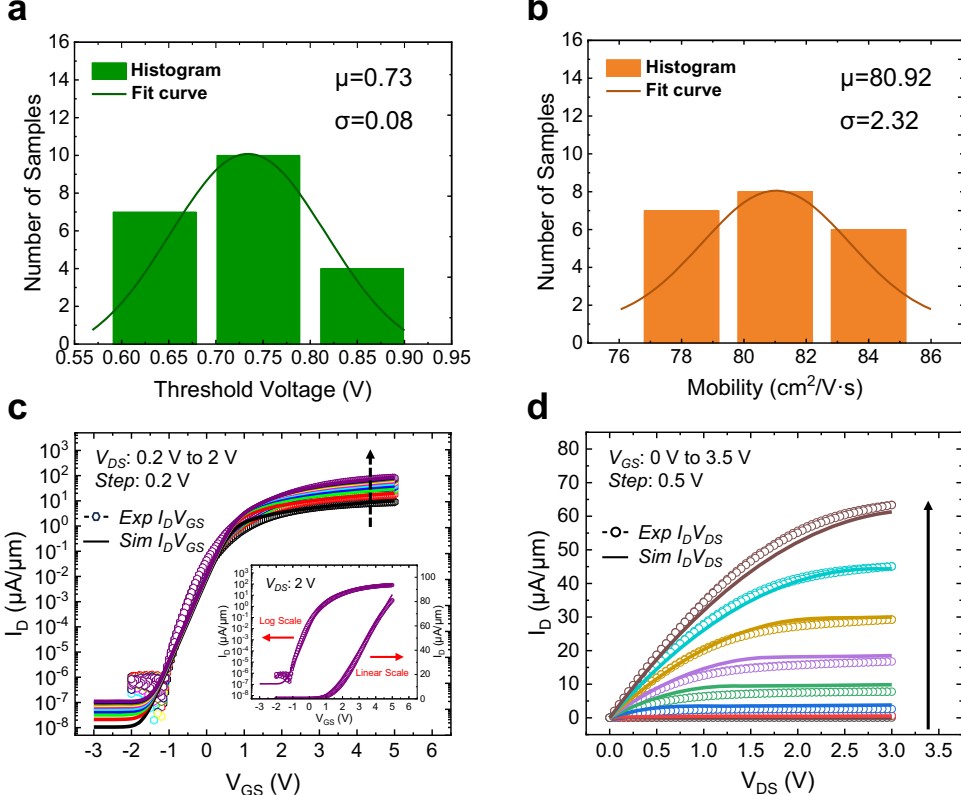

**Fig. 4 | ZnO TFT modeling and simulation. a, b** Extracted statistical characteristics of threshold voltage and mobility from 21 fabricated ZnO TFTs, with the Gaussian fit curves. **c, d** Excellent agreement between the simulation and experimental results for transfer curves and output curves, with an inset in the plot showing the behavior of a fit for just one curve (both log scale and linear scale).

the absence of $R_{Contact}$, while $\mu_{FE}$ represents the effective mobility exhibit by the ZnO TFT in the presence of $R_{Contact}$. Details of the mobility extraction method is shown in Supplementary Note S2. A maximum $\mu_{FE}$ of 85 cm²/V·s and $\mu_o$ of 140 cm²/V·s of the ZnO TFT is achieved for $L_{CH}$ = 5 μm. The fabricated ZnO TFT with HfO₂ passivation layer was also observed to be much more stable over time as compared to the un-passivated one. For the passivated one, time stability over 90 days with negligible degradation was achieved. However, the un-passivated ZnO TFT degraded significantly over time, and ceased to work after one month (Supplementary Fig. S7).

Table 1 highlights the merits of the ALD ZnO TFT in this work with similar recently reported devices, including $I_{ON}/I_{OFF}$, subthreshold swing (SS), $V_{TH}$ and $\mu_{FE}$. A low SS for fast switching, high $I_{ON}/I_{OFF}$ and positive $V_{TH}$ for low leakage current and low-power circuits, and high mobility for large drive current are desired. Especially, the mobility of the ZnO TFT is of particular importance as it determines the amount of driving power it can provide to drive either memory cells for emerging memory-centric computing or functional logic circuits. The optimized ZnO TFT studied in this work exhibits the highest $\mu_{FE}$ as compared to other reported ALD-deposited ZnO TFT, while exhibiting excellent performance all round.

**Modeling and simulation**
In this section, we describe a SPICE-compatible model that is further developed to capture the unique DC characteristics of polycrystalline ZnO TFTs. The main equations to implement the model are shown in Supplementary Table 1. The features of the model are as highlighted: (1) For the above-threshold region, the gate-bias dependent field-effect mobility model has been proposed to account for the grain-boundary induced trap states properly, where the exponent factor $m$ in Eq. (3) (Supplementary Table S1) serves as the field enhancement fitting

parameter for the mobility. An empirical function is also introduced to enable the smooth transition between the linear and saturation region; (2) For the subthreshold region, the current is dominated by the diffusion branch since most of the induced charge is trapped by the deep acceptor states (for n-type devices), with the subthreshold ideality factor $\eta$ indicating the property of gain-boundaries[57]. Furthermore, the expression for the SS can be reduced as Eq. (6) (Supplementary Table S1), from which the density of states for the interface traps ($D_{it}$) can be obtained. From Eq. (6) in Supplementary Table S1, the SS of the TFT is proportionate to the temperature ($T$) and $D_{it}$, and inversely proportionate to the oxide capacitance ($C_{ox}$). In our measurements, the $D_{it}$ can be directly inferred from the SS extracted from the transfer curves of the ZnO TFTs while possessing knowledge on the $C_{ox}$. The interface property between the ZnO channel and HfO₂ dielectric is crucial for the transistor operation. It has been well-established that the existence of $V_O$ at the interface should be treated as the origin of interface states[58,59]. Charge transfer will occur between the channel carriers and interface states, which leads to degraded transport and poorer electrostatic control over the channel. Therefore, low $D_{it}$ is highly desired to enable a transistor with enhancement mode, hysteresis-free, high mobility, and suppressed SS characteristics.

As in the case of IGZO oxide semiconductor, the depletion-mode FET with significant charge-carrier density in the channel at $V_{GS}$ = 0 V is undesirable for applications, since a negative gate-to-source voltage is required to turn off the transistor. However, different from IGZO which is amorphous in nature, the ZnO film studied in this work is polycrystalline, supported by XRD characterization in Fig. 2c, d. This results in improved mobility due to suppressed carrier scattering, indicating the potential of ZnO. In addition, our well-controlled ALD deposition process of ZnO film eliminates the interface dipoles and trap states, which reduces the mobile charges present in the channel significantly.

A nearly charge-neutral ZnO–HfO$_2$ interface can thus be obtained to enable the desirable enhancement-mode characteristics. The $V_{TH}$ and $\mu_{FE}$ are extracted out of 21 fabricated ZnO TFTs, which generally show a gaussian distribution as illustrated in Fig. 4a, b with the mean value and standard deviation values provided. Figure 4c, d shows the excellent agreement between the simulation and experimental results for the transfer and output curves, respectively, after the extracted $\mu_{FE}$, $V_{TH}$, and geometrical parameters are fed into the model. The inset in Fig. 4c shows the behavior of a fit for just one curve (both log scale and linear scale).

### Demonstration of ZnO TFT as memory driver in a 1T1R array

The low processing temperature and high $\mu_{FE}$ make the ZnO TFT suitable to be M3D integrated as a select transistor for driving RRAM in CMOS-BEOL process for memory-centric computing. In this section, we discuss on the analog states tuning functionality and repeatability of the 1T1R memory cell. In the 1T1R array, the gate of the ZnO TFT served as the word line, while the top electrode of the RRAM served as the bit line. The RRAMs were characterized separately first. Figure 5a shows the DC switching characteristics of a single RRAM over 150 cycles (current compliance was set as 1 mA), showing a functional single RRAM device. Multiple RRAMs DC characteristics and the cumulative probability plot of high-resistance state (HRS) and low resistance state (LRS) distribution are shown in Supplementary Fig. S8, demonstrating good uniformity of the fabricated devices. Following, DC switching characteristic of a 1T1R memory cell (applied $V_{GS}$ = 3 V) displays good repeatability and a first-order lower reset current (Fig. 5b). The lower reset current compared to single RRAM is attributed to the impedance and current limited by the ZnO TFT. In the 1T1R measurement, no current compliance was used, and repeatable performance was obtained, albeit with a comparatively larger reset voltage due to the potential drop across the ZnO TFT. Figure 5c shows the cumulative probability plot of $V_{SET}$ and $V_{RESET}$ distribution for the 1T1R memory cell measured over 35 memory cells. The median values of $V_{SET}$ and $V_{RESET}$ are 0.9 V and −1.5 V, respectively.

### Simulation and experimental demonstration of ZnO TFT-based inverter

Based on the calibrated model, three types of inverter configurations, i.e., the pseudo enhancement load (PEL), linear enhancement load (LEL), and depletion load (DL), are evaluated to compare the sensitivity and tolerance of the ZnO gate-level circuit to the device variabilities. The circuit diagrams and corresponding optical images are shown in Fig. 6a–c. The designed inverters are demonstrated experimentally. The voltage transfer curves (VTCs) of the inverters are designed by optimizing the design parameters of the load component (i.e., $W_{MI}$) to ensure a better noise margin for each construct. The $W_{CH}$ and $L_{CH}$ of

load transistors (M1) are 3 μm and 5 μm in PEL and LEL inverters, while 500 μm and 5 μm in DL inverter. The $W_{CH}$ and $L_{CH}$ of other TFTs in inverters are 10 μm and 5 μm, respectively. Figure 6d–f shows the simulated and experimental measured VTCs of the PEL, LEL, DL inverters, with excellent agreement. For the PEL and LEL inverters, the applied $V_{DD}$ is 1.5 V, while $V_{Bias}$ varies from 1.5 to 3 V with step size of 0.5 V. For the DL inverter, the applied $V_{DD}$ varies from 1 V to 3 V with step of 0.5 V. The voltage gain and noise margin (NM) with $V_{DD}$ of 1.5 V of the three different inverters are shown in Supplementary Fig. S9. Of the three different kinds of inverters, the PEL inverters can realize rail-to-rail operation, i.e., switching from $V_{DD}$ to $V_{SS}$ with low-noise margin ($NM_L$) and high noise margin ($NM_H$) of 0.26 V and 0.65 V, respectively. The VTC has a positive shift with the fixed $V_{DD}$ with increasing $V_{Bias}$ due to the increasing $V_{IM}$, leading to impedance reduction of $M_{UN}$. While the DL inverter can also achieve rail-to-rail operation, the designed transistor has to be excessively large to ensure a sufficiently low pseudo-load impedance and is thus less desired. On the other hand, the LEL inverter while functional, cannot realize rail-to-rail operation due to the finite pseudo-load impedance of the pull-up transistor. Further VTCs of the PEL and LEL inverters with different $V_{DD}$ are shown in Supplementary Fig. S10, while the square pulse responses for these three different inverters with input frequency of 1 kHz are shown in Supplementary Fig. S11. We provide a benchmark table of various critical parameters of the three different inverters summarized in Supplementary Table S2. The designed PEL inverter is shown to be the most robust configuration due to the two-stage structure.

### Simulation and experimental demonstration of ZnO TFT-based ring oscillator

To further explore the merits of high-mobility ZnO TFTs at the circuit level, the 5-stage ring oscillators (ROs) based on these three different types of inverters were designed. The 5-stage ROs were verified experimentally; the optical images of the fabricated ROs are shown in Supplementary Fig. S12. Figure 7a, b shows the as-measured and simulated output waveforms of the PEL inverter-based ROs, respectively. The measured frequency of the PEL inverter-based RO is at 369.1 kHz, while the simulated frequency is at 385 kHz. The measured frequency of the LEL inverter-based RO is the highest at 558.2 kHz, and the DL inverter-based RO at 10.3 kHz (Supplementary Fig. S13). The three different ROs performances, including measured frequency, peak-to-peak voltage, working current, and calculated delay time per stage under different applied voltage pairs, are summarized in Supplementary Tables S3–S5.

The measured frequencies of the fabricated ROs are observed to be smaller than that of the simulations because of parasitic resistance and capacitance of the large contact pads, wirings, and the measurement setup. The parasitic resistance of the fabricated ROs is assumed

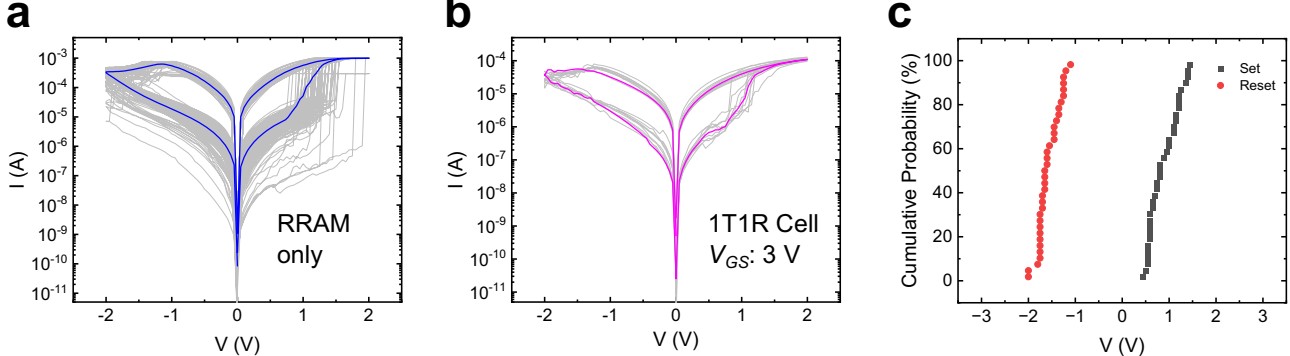

**Fig. 5 | 1T1R electrical performance. a** DC characteristic of a single HfO$_2$ RRAM over 150 switching cycles. The blue line shows the average switching curve. **b** DC characteristics of a 1T1R memory cell over ten switching cycles ($V_{GS}$ = 3 V). The magenta line shows the average switching curve. **c** Cumulative probability plot of $V_{SET}$ and $V_{RESET}$ distribution for the 1T1R memory cell measured over 35 memory cells. The median values of $V_{SET}$ and $V_{RESET}$ are 0.9 V and −1.5 V, respectively.

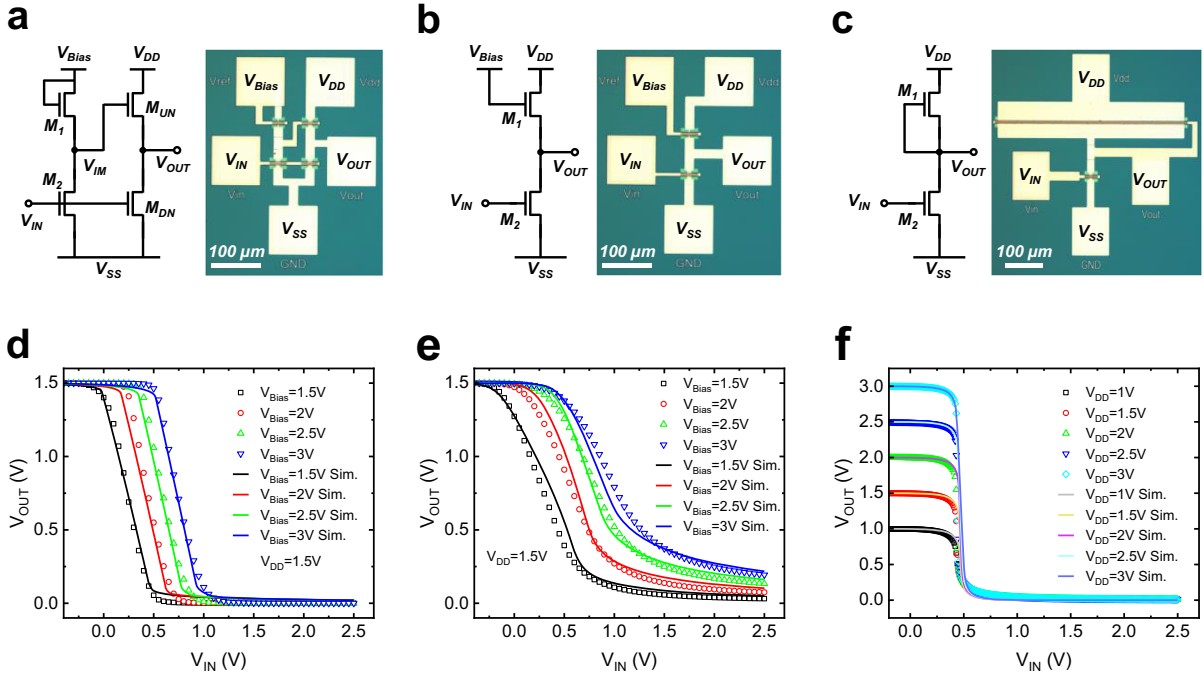

**Fig. 6 | Simulated and experimental voltage transfer curves (VTCs) of inverters. a–c** Circuit diagrams and optical images of the PEL, LEL, and DL inverters. Simulation and experimental measurements of (**d**, **e**) VTC of PEL and LEL inverters, with $V_{DD}$ of 1.5 V and $V_{Bias}$ varying from 1.5 to 3 V and **f** VTC of DL inverter with $V_{DD}$ varying from 1 to 3 V.

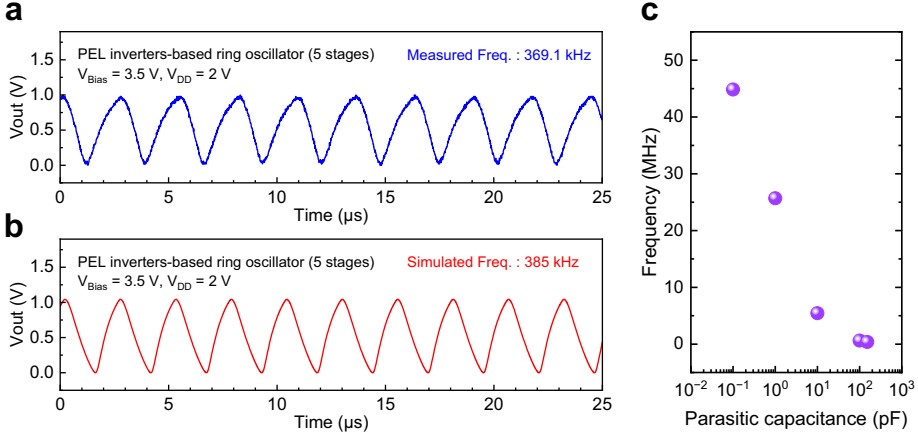

**Fig. 7 | Simulations and experiments of PEL inverter-based ring oscillator. a, b** The as-measured and simulated output waveforms for the 5-stage PEL inverter-based ring oscillator. **c** Projected frequency (simulated) versus parasitic capacitance of the RO circuits.

to be negligible due to the size of the wirings designed. However, the parasitic capacitances are non-trivial and should be accounted for. The total capacitance of PEL inverter-based RO is estimated to be ~150 pF, with details provided in the methods section. With the inclusion of the parasitic capacitance in simulation, both frequencies of the ROs from simulation and experiment are matched as expected. In actual circuit design, the parasitic capacitances are usually minimized without unnecessary large contact pads and are typically less than 0.1 pF. With a typical parasitic capacitance of 0.1 pF, the frequency of the RO can easily reach 10's of MHz as projected using simulation, owing to the high mobility achieved of the proposed ZnO TFT. Figure 7c shows the simulated frequency versus parasitic capacitance, where the frequency can reach up to 44.85 MHz with parasitic capacitance of 0.1 pF. Hence, it is evident that the designed ROs can perform equally well as

compared to the simulated results; the oscillating frequencies of the ROs can be remarkably improved by reducing wiring areas and upgrading the measurement systems. Our results, spanning from device process optimization, device compact modeling, system integration, and circuits designs (both simulations and experiments), demonstrate the potential and suitability of ZnO TFT to be implemented as circuit-level devices in CMOS BEOL.

In summary, we have reported on a CMOS-BEOL-compatible, temperature optimized, low-temperature (200 °C) fabricated ALD ZnO TFT with excellent $\mu_{FE}/\mu_o$ of 85/140 cm$^2$/V·s, high $I_{ON/OFF} > 10^8$, low $SS$ of 110 mV/dec, $D_{it}$ of $2.45 \times 10^{11}$ eV$^{-1}$ cm$^{-2}$, and negligible hysteresis <50 mV. In order to co-design for future M3D architecture requiring BEOL integration, we have further evaluated the implementation of our ZnO TFT in 1T1R memory array as well as functional logic

circuitries and ROs. These were achieved using both novel compact/spice modeling and experimental verification. We have shown that a PEL type inverter configuration provides the most robust performance with excellent noise margin. Finally, a 5-stage PEL inverter-based RO design can achieve a working frequency reaching 369.1 kHz, and projected to reach 10's of MHz in the absence of undesired parasitic capacitance as verified from both simulations and experiments. Hence, our results pave the way for potential implementation of ZnO TFT-based circuitries in future M3D computing systems.

## Methods

### Device fabrication

The fabrication process started with the standard cleaning steps with the SC-1 and SC-2. The bottom-gate electrode composed of Ti/Pt (the thickness is 5/23 nm) was first deposited by e-beam evaporation (EBE) onto a 285 nm $SiO_2$ layer on a Si substrate. The evaporation vacuum value of EBE was less than 5E-6 torr. The deposition rates for Ti and Pt are 4 Å/s and 2 Å/s, respectively. Then the 100 cycles $HfO_2$ layer (10-nm thick, measured by ellipsometer) was deposited as gate dielectric by ALD using tetrakis (ethylmethylamido) hafnium (TEMAH) as precursor and $H_2O$ as reactant at 250 °C, while the 100 cycles ZnO active channel layer (15 nm thick) was deposited by ALD using diethylzinc (DEZ) as precursor and $H_2O$ as reactant at three different temperatures—150/200/250 °C for process optimization. In $HfO_2$ ALD process, the pulse time and purge time for TEMAH are 1.6 s and 10 s, while for $H_2O$ are 0.1 s and 10 s, respectively. The carrier gas flows for TEMAH and $H_2O$ are 80 sccm and 100 sccm, respectively. In ZnO ALD process, the pulse time and purge time for DEZ are 0.1 s and 4 s, while for $H_2O$ are 0.2 s and 4 s, respectively. The carrier gas flows for DEZ and $H_2O$ are 150 sccm and 200 sccm, respectively. The TEMAH source bottle is heated to 120 °C, while DEZ source is kept at room temperature during the deposition process. Both the gate dielectric and ZnO channel regions were defined via standard lithography followed by a buffered oxide etch (BOE). The BOE etching rates for $HfO_2$ and ZnO layers are about 0.1 nm/s and 2 nm/s through calibration, respectively. Then, the source/drain electrodes (Ti/Pt – 5/23 nm) were deposited using EBE followed by a lift-off process (same as a bottom gate). After that, a 5-nm-thick $HfO_2$ layer was deposited using ALD to passivate the ZnO TFT, and the deposition recipe is the same as before. Finally, open pad was conducted through standard lithography followed by BOE etching. ZnO TFTs with channel widths ($W_{CH}$) of 10 µm, and channel lengths ($L_{CH}$) of 2/5/10 µm were fabricated. In the standard lithography process, AZ5214 photoresist and RZX3038 developer are used to pattern. AZ5214 does not attack the ZnO film and protects the ZnO channel region during the developing process. The developer contains TMAH, 2.38%. As TMAH can attack ZnO film, during the source/drain photolithography process, we finely and accurately control the developing time to alleviate the damage to ZnO. When the developing time is 38 s, the photoresist can be completely removed, while maintaining negligible damage to the ZnO film.

### Characterization and measurement

All film thicknesses were measured by ellipsometer TF-UVISEL. Scanning electron microscope (SEM) observations were performed with Hitachi SU8320. Transmission electron microscope (TEM) observations were performed with Hitachi HT7700 Exalens operating at 120 kV. Cross-sectional TEM specimen was prepared by focused ion beam (FIB) using Helios NanoLab 600i apparatus. Atomic force microscope (AFM) observations were performed with Dimension Edge. X-ray diffraction (XRD) measurements were performed with Rigaku Smartlab. X-ray photoelectron spectroscopy (XPS) measurements were performed with Thermo Fisher Escalab Xi +. Current–voltage ($I$–$V$) characteristics of ZnO TFTs, capacitance–voltage ($C$–$V$) of $HfO_2$ dielectric MIM capacitors, RRAM and voltage transfer characteristics (VTCs) of all inverters were measured with an Keithley 4200-SCS

semiconductor parameter analyzer under room temperature and ambient conditions. In both the transfer curve and output curve measurement, the sweep rate was 14 points per second. The square pulses were generated by Tektronix AFG1022. The oscillation waveforms of the ring oscillator were measured by Tektronix TBS1102B.

### Device and circuit simulation

The calibrated SPICE-compatible model is further used for the circuit design and simulation with the Specter simulator. For the inverter simulation, the geometry parameters R= $(W/L)_{load}/(W/L)_{driver}$, as well as the additional bias voltage sources, are elaborately designed and simulated iteratively to optimize the characteristics of VTCs for different configurations (PEL, LEL, and DL), which serves as guidelines for fabrication and eventually agrees well with measured results. Subsequently, 5-stage well-optimized inverters are further cascaded to implement the ROs, with distributed parasitic capacitances from the wirings ($C_{wire}$~6 pF per stage for the PEL type, ~30 pF for 5 stages), oscilloscope ($C_{oscilloscope}$~20 pF), and measurement systems ($C_{meas}$~102 pF) are introduced as the load at the output of each corresponding stage.

## Data availability

All data are available from the corresponding authors upon reasonable request.

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

## Acknowledgements

This work was supported by the National Natural Science Foundation of China (Grant No. 62174074—Y.L., 62274081—L.L., 52273246—F.Z.), Shenzhen Fundamental Research Program (Grant No. JCYJ20220530115014032—Y.L., JCYJ20220530115204009—F.Z.), Young Innovative Talent Project Research Program (Grant No. 2021KQNCX077—F.Z.), Zhujiang Young Talent Program (Grant No. 2021QN02X362—Y.L.), Guangdong Provincial Department of Education Innovation Team Program (2021KCXTD012—Y.L.), Special Funds for the Cultivation of Guangdong College Students' Scientific and Technological Innovation (Grant pdjh2022b0455—W.W., pdjh2023c11507—J.L.), SUSTech SME-Pixelcore Neuromorphic In-sensor Computing Joint Lab and Guangdong Provincial Engineering Research Center of 3-D Integration. We would also like to acknowledge the Core Research Facilities (CRF) at SUSTech for the facilities used, and the technical support provided by the staff and engineers at the CRF.

## Author contributions

Y.L., P.Z., and L.L. conceived the concept, and designed the experiments. W.W. and J.L. fabricated the devices. W.W., J.L., and M.S. performed electrical and material characterizations. P.Z. and K.L. performed the compact modeling and simulation. Y.L., P.Z., L.L., and W.W. analyzed the results and co-wrote the manuscript. Z.W., X.F., M.S., H.Y., K.C., J.L., and F.Z. contributed toward data analysis and revision of the manuscript.

## Competing interests

The authors declare no competing interests.
