## [Peer Review File · Nature Communications]

REVIEWER COMMENTS

Reviewer #1 (Remarks to the Author):

The manuscript titled 'CMOS Backend-of-Line Compatible Memory Array and Logic Circuitries Enabled by High-Performance Atomic Layer Deposited ZnO Thin-Film Transistor' is very interesting, well-written, and addresses a technologically important issue due to the growing demand for data-driven processing applications. The “dorsal spine” of the manuscript lies in the study of ZnO thin film transistors manufactured by Atomic Layer Deposition (ALD). The noteworthy results include a comprehensive analysis, ranging from single-device assessment to circuit-level operational evaluation, such as ring oscillators. I strongly believe that this manuscript would serve as a valuable reference for researchers working on CMOS BEOL devices. The manuscript's findings exhibit a significant advancement in the state of the art for ALD ZnO TFTs. The majority of conclusions and claims are well-supported by the results. I have provided suggestions for several points that I consider important for improvement. The methodology employed in the study is highly appropriate, aligning with the standard approach of semiconductor device researchers. Additionally, the paper provides sufficient details to facilitate replication by other researchers. However, in order to be suitable for publication in Nature Communications, I believe that further enhancements and refinements are necessary in several areas, as suggested below.

Suggestions to improve the paper.

1. Line 43. It seems to me that ref. 11 is not suitable to appear at this point.
2. In Lines 74-76, in the period “Excellent electrical properties, including low positive threshold voltage (V_{TH} , 0.72 V), negligible hysteresis (<50 mV), and low Dit (2.45×10^{11} eV⁻¹cm⁻²), were also achieved.” is mentioned about and Dit. This is an important parameter cited in the text, including the abstract and conclusion, to support the claim of excellent properties of the manufactured TFTs. However, in the manuscript, there is no appropriate analysis of the density of states of interface traps. It is necessary to add Dit analyses based on the data presented in a manner compatible with the importance given to this information.
3. Line 98. In the phrase “15 nm thick ZnO active channel layer was then deposited by ALD at 3 different temperatures -150/200/250 °C, where the effect of the deposition temperature is discussed later.”, the word “where” is not used correctly.
4. Fig. 2d. This figure shows the average crystallite size of the ZnO films. It is unclear whether these results were obtained from several samples or just from a typical one. In any case, this information is lacking, and is strongly recommended to add error bars to the graph.
5. Fig. 2f. These data need statistical analysis, including error bars in the plots to support the evaluation of the superiority of 200°C ZnO film deposition by containing the largest amount of oxygen vacancies (V_o).

6. Line 131. Here, reference is made to AFM images from Supplementary Figure S3. I suggest looking carefully to see if there are any artifacts in the AFM images that could be eventually originate from tip faults. I make this suggestion because the image appears slightly distorted, showing a the typical signature of the tips problem.

7. Line 154. The statement "suggesting the detrimental effect of higher trap states." It is vague, not supported by results, and no one reference is cited.

8. Line 155. The explanation "On the other hand, the existing hump in the 250 °C deposited ZnO TFT suggests the undesired formation of a 2nd channel due to the creation of non-uniform layers during the deposition process" is a statement that requires a convincing argument to support it. Additionally, it is not mentioned whether this curve illustrates a typical behavior or if it is observed in the same manner for all the samples.

9. Line 157. It is informed that "The 200 °C deposited ZnO TFT shows the best overall performance with the highest drive current and a low positive V_{TH} , and is further discussed." In addition to this information, for clarity, it would be beneficial to add that Figure 3c shows transfer curves from a TFT using ZnO deposited at 220°C.

10. Line 158 It is stated that "Fig. 3c shows the normalized transfer curves...". In fact, all output and transfer curves shown in the manuscript are normalized. It is necessary to add the information that the current I_{DS} is normalized to the width of the channel.

11. Fig. 3a. In the vertical scale, is the unit mA/ μm correct?

12. Fig. 3c e 3d. The legend should provide information about the insets.

13. Line 162 and legend of Fig.3. What scan rate is used for the sweep of V_G in transfer or V_{DS} in output curves? This information is not provided anywhere.

14. Fig 3c and 3d. Both insets do not clearly show the hysteresis. In Figure 3d, it appears that the colors of the inset curves do not match the colors used in the main graphics. It is necessary to consider how to visually represent the small hysteresis. At some points, the hysteresis is referred to as "< 50mV". It is important to specify somewhere how the hysteresis is measured.

15. Line 165. It is stated that "Furthermore, the fabricated ZnO TFTs show good uniformity, as can be seen from the transfer curves of 20 devices randomly selected across entire sample with minor variations ($\Delta V_{TH} = 0.22$ V), as shown in Supplementary Fig. S4." I suggest presenting the main parameters as V_{th} , I_{on}/I_{of} , μ , SS e D_{it} obtained for all samples in a table along with their statistical values (at least mean values and standard deviations). This will provide a better representation of the variability between the devices.

16. Line 167. The author could enhance the comments regarding the significance of the contact resistance. This experimental result plays a crucial role in obtaining intrinsic mobility (μ_0), which is highly important. Additionally, there seems to be a misunderstanding regarding the terms "field effect mobility" (μ_{FE}) and "intrinsic mobility" (μ_0) in the manuscript. It appears that μ_0 represents the true field effect mobility as it accounts the impact or error induced by contact resistance. This is clearly described in Equation 2 in Supplementary Note S2. Therefore, it is necessary to accurately specify the term "intrinsic mobility" (μ_0) and carefully review both citations, for accuracy, in all manuscript.

17. Line 170. The statement “A total RContact of 0.99 kΩ·μm was extracted at the fitting line’s vertical intercept point.” does not align with the units or the data presented in Figure 3b, nor with the equations or data insets. It is necessary to ensure consistency between the insets, the figure data, and the textual information.

18. Line 176. It is stated that “Time stability over 90 days (passivated) with negligible degradation in performance as compared to that less than a week (un-passivated) was achieved, as reported in our previous work⁴⁹.” I understand that this information is crucial because it provides a very important characteristic of ZnO TFT obtained with the HfO₂ layer passivation. This characteristic is emphasized at various points in the manuscript, from the abstract to the conclusion. Therefore, I believe that this topic deserves further attention. It would be beneficial to supplement the manuscript with additional data in the form of a graph or table, illustrating the evolution of parameters over time with aging. Although there is a self-citation that addresses this subject, it should be noted that the referenced paper is a conference paper, which may not be easily accessible and does not explore the theme with the importance required in the current manuscript.

19. Table 1 - first comment. This table is shown data about; i. processing temperature; ii. ZnO thickness; iii. insulate(oxide) thickness, iv. Ion/Ion ratio; v. subthreshold swing, SS; vi. threshold voltage, VTH; vii. field effect mobility, μFE; and viii. intrinsic mobility, μo. But despite the comparison of essential parameters in the table, the text only includes a brief comment on mobility. It is necessary to align the manuscript's text with the information in the table by providing additional comments on the other parameters. In this context, comments on parameters such as SS and VTH are particularly important.

20. Table 1 - second comment. In the last column, the saturation mobility (μsat) stands out as something different from the other mobility values used in the manuscript (field effect mobility and intrinsic mobility). As the author uses the equation cited in Supplementary Note S2 $\mu_{FE} = G_m / (C_{ox} W / L V_{DS})$, looks to me that they are using the saturation mobility too. In this way do not need to specify μsat in the table.

21. Table 1 - third comment. As the main purpose of this table is to compare the overall benchmark ALD ZnO TFT, I recommend putting all the data in each column in the same standard format, including significant figures and decimal places.

22. Line 182. In the period “The optimized ZnO TFT studied in this work exhibits the highest mobility (both μFE and μo) as compared to other reported ALD deposited ZnO TFT” the information in parentheses “both μFE and μo” do not apply. In this case μo do not appear in any reference cited. In fact, there is no need to mention μo in the table because it is only presented in the last line, specifically in the data from this work.

23. Fig. 4. The arrangement of the Fig. 4a is effective in displaying all the data set together. It provides an idea of the fit quality in all experimental ranges of curves. However, it loses effectiveness in showing the behavior of the fit for just one curve. I suggest including inset only one experimental curve (for a specific VDS voltage value) together with its corresponding simulated curves.

24. Fig. 4. In Fig. 4c and 4d the standard deviation and medium value need to be expressed in a format that will be compatible with significant figures.

25. Fig. 4. Is the continuous line in Fig. 4 c e 4d, the graphics of one Gaussian curve? If so, why is it not traced across the entire range of the horizontal axis? This information needs to be clearly stated.

26. I suggest inverting the order of showing the Fig. 4a and 4b with the Fig. 4c and 4d. I am suggesting this because the way it is presented on the manuscript may lead the reader think that the mobilities shown in Fig. 4c and 4d are obtained from the simulations. However, the phrase “Fig. 4a and 4b show the excellent agreement between the simulation and experimental results for the transfer and output curves respectively, after the extracted μ_{FE} , V_{TH} and geometrical parameters are fed into the model.” describes the opposite: the simulation is done using this set of experimental data. Moreover, these data are very good to explain the variability of device to device, and for this reason, will be better to appear before and separated.

27. There are 10 citations of “conference papers” in the references. Some of them seem to be of minor importance and, therefore, dispensable. I suggest reviewing the reference list, paying attention to this point.

Reviewer #2 (Remarks to the Author):

This paper presents a study on high-performance ZnO TFT with CMOS backend-of-line application, including memory array and logic circuitries. However, the paper requires revisions and additional discussions.

The abstract might need to be revised according to the required format. It should include the purpose, issue, results, and advantages of the study.

Introduction

The purpose and issue should be clearly described.

L85-90: References need to be inserted.

Fig. 1: TFT Channel

In Fig. 1c)-1), the gate contact is not shown, which might appears to be unusual.

Detailed condition of ZnO deposition with ALD might be helpful for readers to understand the fabrication process.

Could you please provide information about the substrate material?

How was the ZnO etched in the fabrication process?

ZnO is highly soluble in alkali and acid solutions, making it important to provide the readers with information about the material variability. Additionally, defining Ti/Pt on ZnO can be challenging. The authors might have used a strong alkaline developer on the ZnO thin film which damage the semiconductor too much.

Fig. 2

c: XRD diffraction peaks should be labeled. Why do the XRD patterns differ significantly from each other?

d: The calculation of crystallite size should be described in detail with references. There are likely several assumptions made to calculate the crystallite size.

e: Reference should be added in the caption for XPS results.

Fig. 3a:

What is the reason for the difference observed in the ZnO TFT characteristics with different temperature?

Could you provide information about the thickness employed for the TFT characteristics in Fig. 3a?

What are the parameters used to calculate μ_{FE} , such as relative permittivity?

c: Description about the inserted graph should be added in the caption.

d: Description about the inserted graph should be added in the caption.

Fig. 7:

Are the mobility values consistent between the ring oscillator and TFT characteristics?

The text should include a description of D_{it} .

Consider discussing oxide TFTs such as ZnO TFT, IGZO, and BEOL-IGZO, etc.

What is the reason for the enhanced mode TFT and high μ_{FE} ?

Author's Response to Reviewer's comments

We sincerely thank the reviewers for the insightful and constructive comments to improve the manuscript. We have revised our manuscript to address the questions and concerns raised by the reviewers. Please find the point-by-point response to the reviewer's comment below in **Blue**, and the revised text in manuscript in **Red**. The changes made in the manuscript are highlighted in **yellow** for reference.

REVIEWER COMMENTS

Reviewer #1 (Remarks to the Author):

The manuscript titled 'CMOS Backend-of-Line Compatible Memory Array and Logic Circuitries Enabled by High-Performance Atomic Layer Deposited ZnO Thin-Film Transistor' is very interesting, well-written, and addresses a technologically important issue due to the growing demand for data-driven processing applications. The “dorsal spine” of the manuscript lies in the study of ZnO thin film transistors manufactured by Atomic Layer Deposition (ALD). The noteworthy results include a comprehensive analysis, ranging from single-device assessment to circuit-level operational evaluation, such as ring oscillators. I strongly believe that this manuscript would serve as a valuable reference for researchers working on CMOS BEOL devices. The manuscript's findings exhibit a significant advancement in the state of the art for ALD ZnO TFTs. The majority of conclusions and claims are well-supported by the results. I have provided suggestions for several points that I consider important for improvement. The methodology employed in the study is highly appropriate, aligning with the standard approach of semiconductor device researchers. Additionally, the paper provides sufficient details to facilitate replication by other researchers. However, in order to be suitable for publication in Nature Communications, I believe that further enhancements and refinements are necessary in several areas, as suggested below. Suggestions to improve the paper.

1. Line 43. It seems to me that ref. 11 is not suitable to appear at this point.

We have removed ref. 11 from the manuscript as well as from line 43, and re-organized the reference numbering.

Change in manuscript: **Yes**

Location of Change:

Page 1 (**Introduction**)

“Beyond-Si devices that can be co-integrated additively on Si-based complementary metal-oxide-semiconductor (CMOS) chips, include carbon nanotube (CNT) field-effect transistors (FETs)⁴⁻⁶, two-dimensional (2D) materials⁷⁻¹⁰, and oxide semiconductors^{11,12}.”

2. In Lines 74-76, in the period “Excellent electrical properties, including low positive threshold voltage (V_{TH} , 0.72 V), negligible hysteresis (<50 mV), and low D_{it} ($2.45 \times 10^{11} \text{ eV}^{-1} \text{ cm}^{-2}$), were also achieved.” is mentioned about and D_{it} . This is an important parameter cited in the text, including the abstract and conclusion, to support the claim of excellent properties of the manufactured TFTs. However, in the manuscript, there is no appropriate analysis of the density of states of interface traps. It is necessary to add D_{it} analyses based on the data presented in a manner compatible with the importance given to this information.

Thank you for the suggestion. As presented in Eq. 6 in our supplementary Table S1

$$SS \approx \frac{kT}{q} \ln 10 \cdot (1 + qD_{it}/C_{ox}) \text{ --- (Eq. 6)}$$

, the SS of the TFT is proportionate to the temperature (T) and D_{it} , and inversely proportionate to the oxide capacitance (C_{ox}). In our measurements, the D_{it} can be directly inferred from the SS extracted from the transfer curves of the ZnO TFTs while possessing knowledge on the C_{ox} . The interface property between the ZnO channel and HfO₂ dielectric is crucial for the transistor operation. It has been well-established that the existence of oxygen vacancy (V_O) at the interface should be treated as the origin of interface states. Charge transfer will occur between the channel carriers and interface states, which leads to degraded transport and poorer electrostatic control over the channel. Therefore, low density of states of interface traps (D_{it}) is highly desired to enable a transistor with enhancement-mode, hysteresis-free, high mobility and suppressed SS characteristics. We have included more discussion in the revised manuscript for better clarity.

Change in manuscript: **Yes**

Location of Change:

Page 10 (**Results and Discussion – Modelling and simulation**)

“Furthermore, the expression for the subthreshold swing (SS) can be reduced as **Eq. 6 (Supplementary Table S1)**, from which the density of states for the interface traps (D_{it}) can be obtained. From Eq. 6 in supplementary Table S1, the SS of the TFT is proportionate to the temperature (T) and D_{it} , and inversely proportionate to the oxide capacitance (C_{ox}). In our measurements, the D_{it} can be directly inferred from the SS extracted from the transfer curves of the ZnO TFTs while possessing knowledge on the C_{ox} . The interface property between the ZnO channel and HfO₂ dielectric is crucial for the transistor operation. It has been well-established that the existence of V_O at the interface should be treated as the origin of interface states. Charge transfer will occur between the channel carriers and interface states, which leads to degraded transport and poorer electrostatic control over the channel. Therefore, low density of states of interface traps (D_{it}) is highly desired to enable a transistor with enhancement-mode, hysteresis-free, high mobility and suppressed SS characteristics.”

3. Line 98. In the phrase “15 nm thick ZnO active channel layer was then deposited by ALD at 3 different temperatures -150/200/250 °C, where the effect of the deposition temperature is discussed later.”, the word “where” is not used correctly.

Thank you for pointing it out, we have revised the sentence.

Change in manuscript: **Yes**

Location of Change:

Page 4 (**Results and Discussion – Fabrication of TFT and circuits**)

“15 nm thick ZnO active channel layer was then deposited by ALD at 3 different temperatures -150/200/250 °C. The effect of the deposition temperatures on the TFT performance is discussed later.”

4. Fig. 2d. This figure shows the average crystallite size of the ZnO films. It is unclear whether these results were obtained from several samples or just from a typical one. In

any case, this information is lacking, and is strongly recommended to add error bars to the graph.

We obtained the XRD spectra from 3 samples for each deposition temperature, and have extracted the standard deviation for the average crystallite size. We have revised Fig. 2d to include the error bar.

We have also supplemented the analysis of XRD patterns.

Change in manuscript: **Yes**

Location of Change:

Page 5 (**Fig. 2c, d**)

Fig. 2 | Material characterizations of ZnO channel layer. ... c, d GI-XRD patterns, and average crystallite sizes calculated from the Scherrer's formula respectively, of the 3 different temperatures deposited ZnO films.

Change in manuscript: **Yes**

Location of Change:

Page 5 (**Results and Discussion – Materials characterization**)

“To analyze the crystallinity and the preferred orientation, grazing incidence X-ray diffraction (GI-XRD) scans were obtained for the ZnO thin- films deposited at the 3 different temperatures, as shown in **Fig. 2c**. The XRD patterns show that all the ZnO films are polycrystalline with a hexagonal wurtzite structure, with the peaks identified as (100), (002), (101), (102), (110), (103), (112) phases^{39,40}. The XRD patterns of all the three films exhibit the enhanced intensities for the peaks corresponding to (002) plane, indicating preferential orientation along the c-axis^{40,41}. The 150°C deposited ZnO film shows the preferable (002) orientation with additional (100) and (101) peaks. However, the (100) and (101) peaks weaken significantly, and the (002) peak dominates as the deposition temperature of ZnO films increases. This can be attributed to the low surface energy of (002) plane, which is the most thermodynamically favorable^{40,42,43}. When the deposition temperature of film is high, Zn and O atoms can obtain sufficient energy to transfer themselves into energetically favorable positions, which results in ZnO thin films transiting to the (002) preferred orientation at high deposition temperature⁴⁴⁻⁴⁶. **Fig. 2d** shows the average crystallite sizes of the 3 different ZnO thin-films, which were calculated using the Scherrer's formula (see **Supplementary Note S1** for details). The largest average crystallite size of 5.14 nm is obtained of the 200°C deposited ZnO film, which is favorable for high mobility due to reduced grain boundary

scattering^{47.}”

5. Fig. 2f. These data need statistical analysis, including error bars in the plots to support the evaluation of the superiority of 200°C ZnO film deposition by containing the largest amount of oxygen vacancies (V_o).

We performed 3 independent XPS spectra measurements for each temperature, and the standard deviation from each component extracted. Fig. 2f shows the average atomic ratio of each component. We have added the error bars and revised Fig. 2f.

Change in manuscript: **Yes**

Location of Change:

Page 5 (Fig. 2f)

Fig. 2 | Material characterizations of ZnO channel layer. ... e, f O1s XPS spectra of the ZnO channel layers deposited at 3 temperatures as indicated, and plot showing the Zn-O, V_o , and -OH atomic percentages as a function of the 3 different temperatures used.

6. Line 131. Here, reference is made to AFM images from Supplementary Figure S3. I suggest looking carefully to see if there are any artifacts in the AFM images that could be eventually originate from tip faults. I make this suggestion because the image appears slightly distorted, showing a typical signature of the tips problem.

Thank you for pointing this out, and we have re-scanned the films. In order to present the uniformity of the deposited film more accurately, we increased the scanning range from $1 \mu\text{m} \times 1 \mu\text{m}$ (previously) to $5 \mu\text{m} \times 5 \mu\text{m}$, as well as three independent measurements. The average R_q of HfO_2 on gate and ZnO on HfO_2 dielectric on gate is 0.289 nm (standard deviation: 0.006 nm) and 0.342 nm (standard deviation: 0.013 nm), respectively. We have revised the manuscript with the new measurements.

Change in manuscript: **Yes**

Location of Change:

Supplementary Figure S3 | AFM images (Supplementary Information)

Supplementary Figure S3 | AFM images. a, b AFM images of the HfO₂ dielectric on gate electrode, and ZnO on HfO₂ dielectric on gate during fabrication. The average R_q of HfO₂ on gate and ZnO on HfO₂ dielectric on gate is 0.289 nm (standard deviation: 0.006 nm, over 3 independent scans) and 0.342 nm (standard deviation: 0.013 nm, over 3 independent scans), respectively.

7. Line 154. The statement “suggesting the detrimental effect of higher trap states.” It is vague, not supported by results, and no one reference is cited.

Thank you for pointing it out, trap states are known to detrimentally affect the performance of transistors by reducing the field-effect mobility (μ_{FE}) as well as an undesired over shift in the V_{TH} due to the presence of unstable stray charges that could enter or leave the trap states. This phenomenon has been reported in literature. We have revised the manuscript and include appropriate references to further explain this observation.

Added references

- Zhang, L., Li, J., Zhang, X. W., Jiang, X. Y. & Zhang, Z. L. High performance ZnO-thin-film transistor with Ta2O5 dielectrics fabricated at room temperature. *Applied Physics Letters* **95** (2009). <https://doi.org:10.1063/1.3206917>
- Li, H. *et al.* High-Performance ZnO Thin-Film Transistors Prepared by Atomic Layer Deposition. *IEEE Transactions on Electron Devices* **66**, 2965-2970 (2019). <https://doi.org:10.1109/ted.2019.2915625>
- Yang, J. *et al.* Characteristics of ALD-ZnO Thin Film Transistor Using H₂O and H₂O₂ as Oxygen Sources. *Advanced Materials Interfaces* **9** (2022). <https://doi.org:10.1002/admi.202101953>

Change in manuscript: **Yes**

Location of Change:

Page 7 (**Results and Discussion – ZnO TFT electrical characterization**)

“The 150 °C deposited ZnO TFT exhibits the lowest drive current, lowest μ_{FE} (3.63 cm²/V·s), and largest V_{TH} (2.5 V) as compared to the TFTs fabricated at higher temperatures. This suggests the detrimental effect of higher trap states, which allows unstable stray charges entering or leaving the traps, resulting in reduced μ_{FE} and overly positive shift in the V_{TH} ^{40,45,51}.”

8. Line 155. The explanation "On the other hand, the existing hump in the 250 °C

deposited ZnO TFT suggests the undesired formation of a 2nd channel due to the creation of non-uniform layers during the deposition process" is a statement that requires a convincing argument to support it. Additionally, it is not mentioned whether this curve illustrates a typical behavior or if it is observed in the same manner for all the samples.

The observation of the hump in the transfer curve is typical of the ZnO TFTs fabricated at that temperature. The existence of the hump has been reported in several literature where in unoptimized growth conditions, results in the formation of dual conduction channels in the channel layer, i.e., a main channel and a parasitic channel that turns on at different voltage. We have revised the manuscript to include more explanation and appropriate references for better clarity.

Added references:

- Yang, J. *et al.* Investigation of a Hump Phenomenon in Back-Channel-Etched Amorphous In-Ga-Zn-O Thin-Film Transistors Under Negative Bias Stress. *IEEE Electron Device Letters* **38**, 592-595 (2017). <https://doi.org:10.1109/led.2017.2686898>
- Kim, W.-S. *et al.* Abnormal behavior with hump characteristics in current stressed a-InGaZnO thin film transistors. *Solid State Electron* **137**, 22-28 (2017). <https://doi.org:10.1016/j.sse.2017.08.001>
- Teng, T., Hu, C.-F., Qu, X.-P. & Wang, M. Investigation of the anomalous hump phenomenon in amorphous InGaZnO thin-film transistors. *Solid State Electron* **170** (2020). <https://doi.org:10.1016/j.sse.2020.107814>
- Li, Q. *et al.* Structural Engineering Effects on Hump Characteristics of ZnO/InSnO Heterojunction Thin-Film Transistors. *Nanomaterials (Basel)* **12** (2022). <https://doi.org:10.3390/nano12071167>

Change in manuscript: **Yes**

Location of Change:

Page 7 (**Results and Discussion – ZnO TFT electrical characterization**)

“On the other hand, the ZnO TFTs deposited at 220°C shows the emergence of a small hump in the subthreshold region of the transfer curve (**Supplementary Figure S4**), while a distinct hump phenomenon appears in the 250 °C deposited ZnO TFT that is accompanied by a negative shift of V_{TH} , observable in all devices fabricated at this temperature. This phenomenon has been examined in several works, and can be attributed to the existence of dual conduction channels, i.e., a main channel and a parasitic channel that turn on a different voltage as a result of unoptimized growth conditions⁵²⁻⁵⁵.”

9. Line 157. It is informed that “The 200 °C deposited ZnO TFT shows the best overall performance with the highest drive current and a low positive V_{TH} , and is further discussed.” In addition to this information, for clarity, it would be beneficial to add that Figure 3c shows transfer curves from a TFT using ZnO deposited at 220°C.

Thank you for the suggestion. We have fabricated the ZnO TFTs deposited at 220°C, showing the emergence of a small hump in the subthreshold region of the transfer curve.

We have revised the manuscript to provide discussion on the ZnO TFT fabricated at 220°C. As the transfer curve of the 220°C fabricated TFT is almost overlapping with that of the 200°C fabricated TFT, we have included it supplementary **Fig. S4** instead for better clarity.

Change in manuscript: **Yes**

Location of Change:

Page 7 (**Results and Discussion – ZnO TFT electrical characterization**)

“...On the other hand, the ZnO TFTs deposited at 220°C shows the emergence of a small hump in the subthreshold region of the transfer curve (**Supplementary Fig. S4**).”

Change in manuscript: **Yes**

Location of Change:

Supplementary Figure S4 | Transfer curve of 220°C deposited ZnO TFT (Supplementary Information)

Supplementary Figure S4 | Transfer curve of 220°C deposited ZnO TFT. The transfer curve of 220°C deposited ZnO TFT, with an observable small hump in the subthreshold region (circled dashed line).

10. Line 158 It is stated that "Fig. 3c shows the normalized transfer curves...". In fact, all output and transfer curves shown in the manuscript are normalized. It is necessary to add the information that the current I_{DS} is normalized to the width of the channel.

Thank you for pointing this out, we have revised the statement to reflect that the curves are normalized to the channel width of the TFT (W_{CH}).

Change in manuscript: **Yes**

Location of Change:

Page 7 (**Results and Discussion – ZnO TFT electrical characterization**)

“**Fig. 3a** shows the transfer curves (I_D - V_{GS}) of the ZnO TFTs fabricated at the 3 different temperatures with W_{CH} and L_{CH} of 10 μm and 5 μm respectively. The current-voltage (I - V) curves in Fig. 3 are all normalized to the W_{CH} .”

11. Fig. 3a. In the vertical scale, is the unit $\text{mA}/\mu\text{m}$ correct?

We apologize for the error. The correct unit should be $\mu\text{A}/\mu\text{m}$. We have corrected Fig. 3a to the correct unit.

Change in manuscript: **Yes**
 Location of Change:
 Page 7 (**Fig. 3a**)

Fig. 3 | ZnO TFT electrical performance. a Transfer curves (I_D - V_{GS}) of the ZnO TFTs fabricated at 3 different temperatures (150/200/250°C) with W_{CH} and L_{CH} of 10 μm and 5 μm , respectively.

12. Fig. 3c 3d. The legend should provide information about the insets.

We have revised how we present **Fig. 3c** and **3d** for better clarity on the insets, i.e., circling the region where we did a zoomed-in on the plots. We have also revised the captions accordingly.

Change in manuscript: **Yes**
 Location of Change:
 Page 7 (**Fig. 3**)

Fig. 3 | ZnO TFT electrical performance. ... c I_D - V_{GS} family of curves (dual sweep) measured over different V_{DS} (0.2 to 2V). A high current on-off ratio up to $\sim 10^8$ is obtained with SS of 110 mV/dec and hysteresis < 52 mV. **d** Output (I_D - V_{DS}) family of curves (dual sweep) measured over different V_{GS} (0 to 3.5 V) of the same TFT. The zoomed-in region of the plots in 3c and 3d (circled) are shown in the inset respectively, indicating the small hysteresis measured of our fabricated TFT.

13. Line 162 and legend of Fig.3. What scan rate is used for the sweep of V_G in transfer or V_{DS} in output curves? This information is not provided anywhere.

Our apologies for missing out on that information. In both the transfer curve (Fig. 3c), and output curve (Fig. 3d), the measurement was done with a sweep rate of 14 points

per second. We have included this information in the revised manuscript under “**Methods – Characterization and Measurement**”

Change in manuscript: **Yes**

Location of Change:

Page 14 (**Methods – Characterization and Measurement**)

“In both the transfer curve and output curve measurement, the sweep rate was 14 points per second.”

14. Fig 3c and 3d. Both insets do not clearly show the hysteresis. In Figure 3d, it appears that the colors of the inset curves do not match the colors used in the main graphics. It is necessary to consider how to visually represent the small hysteresis. At some points, the hysteresis is referred to as " $< 50\text{mV}$ ". It is important to specify somewhere how the hysteresis is measured.

Thank you for the suggestion. We have amended Fig. 3d such that the inset curves colors match that of the main graphics. In addition, as indicated in our response to your comment 12, we have circled the region where the curves in the inset are plotted for clarity (Fig. 3c and 3d). The hysteresis is extracted as such: At a selected V_{GS} in the transfer curve, the voltage difference between the forward and backwards sweep was obtained.

In the earlier manuscript, the hysteresis was obtained at the subthreshold region, i.e., at 0.5 V, with extracted value of 47 mV. However, in order to provide more comprehensive analysis of the hysteresis, we have revised the presentation of the hysteresis in the transfer curve (Fig. 3c) by plotting an additional figure in supplementary **Fig. S5** as shown below. In this figure, we show the extracted hysteresis as a function of V_{GS} at V_{DS} of 1 V, together with the definition of hysteresis extraction in the inset. It can be seen that the hysteresis values vary with different V_{GS} ; a maximum hysteresis value of 52 mV is obtained near the saturation voltage of 4.2 V. We have revised the manuscript with the below figure as well as more explanation in how the hysteresis is measured.

Supplementary Figure S5 | Hysteresis Extraction. Extracted hysteresis from Fig. 3c as a function of V_{GS} at a V_{DS} of 1 V, with a maximum hysteresis value of 52 mV. The definition of the hysteresis extraction is shown in the inset

Change in manuscript: **Yes**

Location of Change:

Page 8 (**Results and Discussion – ZnO TFT electrical characterization**)

“The dual sweep I-V characteristics for both transfer and output curves show negligible hysteresis of 47 mV at $V_{GS} = 0.5$ V, as shown in the insets of **Fig. 3c** and **3d**. The hysteresis is extracted as follows: At a selected V_{GS} in the transfer curve, the voltage difference between the forward and backwards sweep was obtained. The extracted hysteresis as a function of V_{GS} at V_{DS} of 1 V as shown in **Supplementary Fig. S5**, with the definition of the hysteresis extraction shown graphically in the inset. The hysteresis values vary with different V_{GS} , and a maximum hysteresis value of 52 mV is obtained near the saturation voltage of 4.2 V. The small hysteresis value of our TFT further suggest the good interface quality with low interface traps.”

Location of Change:

Supplementary Figure S5 | Hysteresis Extraction (Supplementary Information)

Supplementary Figure S5 | Hysteresis Extraction. Extracted hysteresis from Fig. 3c as a function of V_{GS} at a V_{DS} of 1 V, with a maximum hysteresis value of 52 mV. The definition of the hysteresis extraction is shown graphically in the inset.

15. Line 165. It is stated that “Furthermore, the fabricated ZnO TFTs show good uniformity, as can be seen from the transfer curves of 20 devices randomly selected across entire sample with minor variations ($\Delta V_{TH} = 0.22$ V), as shown in Supplementary Fig. S4.” I suggest presenting the main parameters as V_{th} , I_{on}/I_{off} , μ , SS e D_{it} obtained for all samples in a table along with their statistical values (at least mean values and standard deviations). This will provide a better representation of the variability between the devices.

Thank you for pointing this out, we have extracted the important performance parameters of the TFTs with their statistical values (mean values and standard deviations). In order to maintain consistency with the transfer curves presented in main text, we have revised Supplementary Fig. S6, to show transfer curve at $V_D = 0.2$ V.

Change in manuscript: **Yes**

Location of Change:

Page 8 (**Results and Discussion – ZnO TFT electrical characterization**)

“Furthermore, the fabricated ZnO TFTs show good uniformity, as can be seen from the

transfer curves of 20 devices randomly selected across entire sample with minor standard deviations, as shown in **Supplementary Fig. S6.**”

Supplementary Figure S6 | ZnO TFT uniformity. Transfer curves of 20 different ZnO TFTs measured randomly. Similar performance of the devices was verified with minor variations.

Parameter	Mean value	Standard deviation
V_{TH} (V)	0.81	0.09
μ_{FE} ($\text{cm}^2/\text{V}\cdot\text{s}$)	75.89	7.37
SS (mV/dec)	132.8	13.1
I_{on}/I_{off}	1.47×10^8	4.4×10^7
D_{it} ($\text{eV}^{-1}\text{cm}^{-2}$)	3.34×10^{11}	5.8×10^{10}

The mean value and standard deviation of the extracted TFTs’ parameters from **Supplementary Figure S6**

16. Line 167. The author could enhance the comments regarding the significance of the contact resistance. This experimental result plays a crucial role in obtaining intrinsic mobility (μ_o), which is highly important. Additionally, there seems to be a misunderstanding regarding the terms "field effect mobility" (μ_{FE}) and "intrinsic mobility" (μ_o) in the manuscript. It appears that μ_o represents the true field effect mobility as it accounts the impact or error induced by contact resistance. This is clearly described in Equation 2 in Supplementary Note S2. Therefore, it is necessary to accurately specify the term "intrinsic mobility" (μ_o) and carefully review both citations, for accuracy, in all manuscript.

Thank you for the comments. The contact resistance ($R_{Contact}$) is an unwanted parasitic in transistors, where it degrades drive current required for high-speed operation of integrated circuit (IC) chips. This is especially so at advanced technology node where the $R_{Contact}$ can form a significant portion of the entire TFT’s resistance. The intrinsic mobility (μ_o) is defined to be the true mobility of the ZnO TFT in the absence of $R_{Contact}$, while the field-effect mobility (μ_{FE}) represents the effective mobility exhibit by the ZnO

TFT in the presence of $R_{Contact}$. Hence, it is evident that $R_{Contact}$ is desired to be as small as possible in order to uncover the potentials of the channel material. We have revised the manuscript to better emphasize the undesired effect of the $R_{Contact}$, as well as to describe the μ_{FE} and μ_o presented in the manuscript.

Change in manuscript: **Yes**

Location of Change:

Page 8 (**Results and Discussion – ZnO TFT electrical characterization**)

“The contact resistance ($R_{Contact}$) is an unwanted parasitic in transistors, where it degrades drive current required for high-speed operation of integrated circuit (IC) chips. This is especially so at advanced technology node where the $R_{Contact}$ can form a significant portion of the entire TFT’s resistance. Here, we extracted the $R_{Contact}$ of the ZnO TFT via the transfer length method (TLM), by utilizing the different channel length (L_{CH}) TFTs measured and plotting the TFTs’ resistance vs. L_{CH} (3 different channel lengths are utilized here), as shown in **Fig. 3b**.”

, and

“Here, we present on the extraction of both μ_o and μ_{FE} of the fabricated ZnO TFTs. μ_o is defined as the true mobility of the ZnO TFT in the absence of $R_{Contact}$, while μ_{FE} represents the effective mobility exhibit by the ZnO TFT in the presence of $R_{Contact}$. Details of the mobility extraction method is shown in **Supplementary Note S2**”

17. Line 170. The statement “A total $R_{Contact}$ of $0.99 \text{ k}\Omega \cdot \mu\text{m}$ was extracted at the fitting line’s vertical intercept point.” does not align with the units or the data presented in Figure 3b, nor with the equations or data insets. It is necessary to ensure consistency between the insets, the figure data, and the textual information.

Our apologies for the error. The value of $0.99 \text{ k}\Omega \cdot \mu\text{m}$ is supposed to refer to the specific $R_{Contact}$ that is normalized to the W_{CH} of the TFT, while that presented in Fig. 3b is the actual resistance measured of the TFTs. Considering the TFTs’ W_{CH} of $10 \mu\text{m}$ and intercept of 9902Ω , translates to a specific $R_{Contact}$ of $0.99 \text{ k}\Omega \cdot \mu\text{m}$. We have revised the text in the manuscript to reflect on the above and for better clarity.

Fig. 3 | ZnO TFT electrical performance. b TLM measurements of the TFT with 3 different L_{CH} (2, 5, 10 μm), with a maximum standard deviation of 7%. The $R_{Contact}$ is extracted via the vertical intercept as shown in the linear line fit equation; the $2R_{Contact}$ -specific when normalized to the W_{CH} of 10 μm is then obtained as $0.99 \text{ k}\Omega \cdot \mu\text{m}$...

Change in manuscript: **Yes**

Location of Change:

Page 7 (**Fig. 3b**),

and Page 8 (**Results and Discussion – ZnO TFT electrical characterization**)

Fig. 3 | ZnO TFT electrical performance. b TLM measurements of the TFT with 3 different L_{CH} (2, 5, 10 μm), with a maximum standard deviation of 7%. The R_{Contact} is extracted via the vertical intercept as shown in the linear line fit equation; the $2R_{\text{contact-specific}}$ when normalized to the W_{CH} of 10 μm is then obtained as 0.99 $\text{k}\Omega \cdot \mu\text{m}$...

“A total R_{Contact} of 9.902 $\text{k}\Omega$ was extracted at the fitting line’s vertical intercept point, translating to a specific contact resistance $R_{\text{Contact-specific}}$ of 0.99 $\text{k}\Omega \cdot \mu\text{m}$ when normalized to the TFTs’ W_{CH} of 10 μm .”

18. Line 176. It is stated that “Time stability over 90 days (passivated) with negligible degradation in performance as compared to that less than a week (un-passivated) was achieved, as reported in our previous work⁴⁹.” I understand that this information is crucial because it provides a very important characteristic of ZnO TFT obtained with the HfO₂ layer passivation. This characteristic is emphasized at various points in the manuscript, from the abstract to the conclusion. Therefore, I believe that this topic deserves further attention. It would be beneficial to supplement the manuscript with additional data in the form of a graph or table, illustrating the evolution of parameters over time with aging. Although there is a self-citation that addresses this subject, it should be noted that the referenced paper is a conference paper, which may not be easily accessible and does not explore the theme with the importance required in the current manuscript.

Thank you for pointing this out. We have revised the manuscript with the required figures showing the effect of passivation over time in **Supplementary Fig. S7**.

Change in manuscript: **Yes**

Location of Change:

Page 8 (**Results and Discussion – ZnO TFT electrical characterization**)

“For the passivated one, time stability over 90 days with negligible degradation was achieved. However, the un-passivated ZnO TFT degraded significantly over time, and ceased to work after one month (**Supplementary Fig. S7**).”

Change in manuscript: **Yes**

Location of Change:

Supplementary Figure S7 | Passivation effect on performance of ZnO TFT. (Supplementary Information)

Supplementary Figure S7 | Passivation effect on performance of ZnO TFT. a Evolution of transfer curves of the un-passivated ZnO TFT over time with significant degradation. **b** Evolution of transfer curves of the passivated ZnO TFT over time with minor change.

19. Table 1 - first comment. This table is shown data about; i. processing temperature; ii. ZnO thickness; iii. insulate(oxide) thickness, iv. Ion/Ion ratio; v. subthreshold swing, SS; vi. threshold voltage, V_{TH} ; vii. field effect mobility, μ_{FE} ; and viii. intrinsic mobility, μ_o . But despite the comparison of essential parameters in the table, the text only includes a brief comment on mobility. It is necessary to align the manuscript's text with the information in the table by providing additional comments on the other parameters. In this context, comments on parameters such as SS and V_{TH} are particularly important.

Thank you for the comment, we have included description of the other important parameters in the manuscript's text for completeness.

Change in manuscript: **Yes**

Location of Change:

Page 8 (**Results and Discussion – ZnO TFT electrical characterization**)

“**Table 1** highlights the merits of the ALD ZnO TFT in this work with similar recently reported devices including I_{ON}/I_{OFF} , subthreshold swing (SS), threshold voltage (V_{TH}), and μ_{FE} . A low SS for fast switching, high I_{ON}/I_{OFF} and positive V_{TH} for low leakage current and low power circuits, and a high mobility for large drive current are desired. Especially, the mobility of the ZnO TFT is of particular importance as it determines the amount of driving power it can provide to drive either memory cells for emerging memory centric computing or functional logic circuits. The optimized ZnO TFT studied in this work exhibits the highest μ_{FE} as compared to other reported ALD deposited ZnO TFT, while exhibiting excellent performance all round.”

20. Table 1 - second comment. In the last column, the saturation mobility (μ_{sat}) stands out as something different from the other mobility values used in the manuscript (field effect mobility and intrinsic mobility). As the author uses the equation cited in Supplementary Note S2 $\mu_{FE} = G_m / (C_{ox} W/L V_{DS})$, looks to me that they are using the saturation mobility too. In this way do not need to specify μ_{sat} in the table.

As you have suggested, we have removed μ_{sat} from the table comparisons to avoid confusion.

Change in manuscript: **Yes**

Location of Change:

Page 9 (**Table. 1**)

Table. 1 Benchmark of reported ALD ZnO TFT performance parameters.

Ref.	Year	Temp. (°C)	ZnO Thickness (nm)	Oxide/ Thickness (nm)	I _{ON} /I _{OFF} Ratio	SS (mV/dec)	V _{TH} (V)	μ _{FE} (cm ² /V·s)
16	2019	200	33	SiO ₂ / 30	2.8×10 ⁹	127	4.00	7.8
24	2019	200	15	ZrO ₂ / 5	~10 ⁷	69	0.10	36.8
25	2019	100	16	Al ₂ O ₃ / 36	~10 ⁷	320	1.23	38.4
17	2020	300	36	HfO ₂ / 30	1.9×10 ⁷	175	1.10	11.8
26	2020	200	33	Al ₂ O ₃ / 60	4.1×10 ⁹	131	3.24	19.6
27	2020	300	30	Al ₂ O ₃ / 40	~10 ⁷	220	6.00	16.2
28	2020	150	14	Al ₂ O ₃ / 40	~10 ⁸	210	0.14	31.1
29	2021	350	11	SiO ₂ / 90	5.0×10 ⁹	/	18.70	43.2
30	2021	100	25	Al ₂ O ₃ / 30	3.0×10 ⁷	210	/	14.3
31	2021	100	16	Al ₂ O ₃ / 36	~10 ⁸	170	1.05	31.2
32	2022	100	23	Al ₂ O ₃ / 36	~10 ⁸	170	1.00	32.1
33	2022	100	15	HfO ₂ /30+Al ₂ O ₃ /10	~10 ⁷	110	-3.00	55.5
34	2022	400	20	Al ₂ O ₃ / 30	~10 ¹⁰	225	3.23	17.9
11	2022	150	20	Al ₂ O ₃ / 50	2.0×10 ⁷	250	1.41	10.7
35	2022	100	20	Al ₂ O ₃ / 40	4.3×10 ⁹	243	1.13	45.3
36	2022	300	3	Al ₂ O ₃ / 5	~10 ⁸	94	/	84.0
37	2023	200	17	Al ₂ O ₃ / 15	~10 ¹²	75	/	32.8
38	2023	200	20	Al ₂ O ₃ / 20	~10 ⁹	179	2.73	43.8
This work	2023	200	15	HfO₂ / 10	~10⁸	110	0.72	85.0

21. Table 1 - third comment. As the main purpose of this table is to compare the overall benchmark ALD ZnO TFT, I recommend putting all the data in each column in the same standard format, including significant figures and decimal places.

Thank you for pointing out the inadequacies. We have revised the table for consistencies as shown in our response to your comment 20.

22. Line182. In the period “The optimized ZnO TFT studied in this work exhibits the highest mobility (both μ_{FE} and μ_o) as compared to other reported ALD deposited ZnO TFT” the information in parentheses “both μ_{FE} and μ_o” do not apply. In this case μ_o do not appear in any reference cited. In fact, there is no need to mention μ_o in the table because it is only presented in the last line, specifically in the data from this work.

Thank you for pointing out this and we agree with your comment. We have revised the manuscript and removed μ_o from the table comparison, shown in our response to your

comment 20.

Change in manuscript: **Yes**

Location of Change:

Page 9 (**Table. 1**)

, and Page 8 (**Results and Discussion – ZnO TFT electrical characterization**)

“The optimized ZnO TFT studied in this work exhibits the highest μ_{FE} as compared to other reported ALD deposited ZnO TFT, while exhibiting excellent performance all round.”

23. Fig. 4. The arrangement of the Fig. 4a is effective in displaying all the data set together. It provides an idea of the fit quality in all experimental ranges of curves. However, it loses effectiveness in showing the behavior of the fit for just one curve. I suggest including inset only one experimental curve (for a specific VDS voltage value) together with its corresponding simulated curves.

As you have suggested, we have improved Fig. 4c (previously Fig. 4a) as shown below. In the improved figure, we have added an inset in the plot showing the behavior of a fit for just one curve (both log scale and linear scale of transfer curve). We hope this revision makes the presentation much clearer.

Change in manuscript: **Yes**

Location of Change:

Page 10 (**Fig. 4c**)

Fig. 4 | ZnO TFT Modeling and simulation. ... c, d Excellent agreement between the simulation and experimental results for transfer curves and output curves, with an inset in the plot showing the behavior of a fit for just one curve (both log scale and linear scale).

24. Fig. 4. In Fig. 4c and 4d the standard deviation and medium value need to be expressed in a format that will be compatible with significant figures.

We have revised the standard deviation and medium value to one with the same decimal places throughout.

Change in manuscript: **Yes**
Location of Change:
Page 10 (**Fig. 4a, b**)

Fig. 4 | ZnO TFT Modeling and simulation. a, b Extracted statistical characteristics of threshold voltage and mobility from 21 fabricated ZnO TFTs, with the Gaussian fit curves.

25. Fig. 4. Is the continuous line in Fig. 4 c e 4d, the graphics of one Gaussian curve? If so, why is it not traced across the entire range of the horizontal axis? This information needs to be clearly stated.

Our apologies for the error. The previous continuous lines in Fig. 4c and 4d are Log-Normal Distribution curves. We have revised them to the Gaussian curve as shown in our response to your comment 24.

26. I suggest inverting the order of showing the Fig. 4a and 4b with the Fig. 4c and 4d. I am suggesting this because the way it is presented on the manuscript may lead the reader think that the mobilities shown in Fig. 4c and 4d are obtained from the simulations. However, the phrase “Fig. 4a and 4b show the excellent agreement between the simulation and experimental results for the transfer and output curves respectively, after the extracted μ_{FE} , V_{TH} and geometrical parameters are fed into the model.” describes the opposite: the simulation is done using this set of experimental data. Moreover, these data are very good to explain the variability of device to device, and for this reason, will be better to appear before and separated.

Thank you for the valuable suggestion. We have inverted the order of Fig. 4a and 4b with Fig. 4c and 4d to avoid potential confusion.

Change in manuscript: **Yes**
Location of Change:
Page 10 (**Fig. 4**)

Fig. 4 | ZnO TFT Modeling and simulation. a, b Extracted statistical characteristics of threshold voltage and mobility from 21 fabricated ZnO TFTs, with the Gaussian fit curves. **c, d** Excellent agreement between the simulation and experimental results for transfer curves and output curves, with an inset in the plot showing the behavior of a fit for just one curve (both log scale and linear scale).

27. There are 10 citations of “conference papers” in the references. Some of them seem to be of minor importance and, therefore, dispensable. I suggest reviewing the reference list, paying attention to this point.

Thank you for pointing this out, we have revised our reference list and remove those that are of minor importance.

References removed:

- Li, Q. et al. in 2022 International Electron Devices Meeting (IEDM) 2.7.1-2.7.4 (2022).
- Subhechha, S. et al. in 2022 IEEE Symposium on VLSI Technology and Circuits (VLSI 390 Technology and Circuits) 292-293 (2022).
- Liao, P. Y. et al. in 2022 International Electron Devices Meeting (IEDM) 12.14.11-12.14.14 (2022).
- Hu, Y. et al. in 2022 International Electron Devices Meeting (IEDM) 8.5.1-8.5.4 (2022).
- Tang, W. et al. in 2022 International Electron Devices Meeting (IEDM) 483-486 (2022).
- Huang, K. et al. in 2022 IEEE Symposium on VLSI Technology and Circuits (VLSI Technology 442 and Circuits) 296-297 (2022).

Reviewer #2 (Remarks to the Author):

This paper presents a study on high-performance ZnO TFT with CMOS backend-of-line application, including memory array and logic circuitries. However, the paper requires revisions and additional discussions.

1. The abstract might need to be revised according to the required format. It should include the purpose, issue, results, and advantages of the study.

Thank you for pointing this out. We have revised the abstract to better address the inadequacies.

Change in manuscript: **Yes**

Location of Change:

Page 1 (**Abstract**)

“The development of high-performance oxide-based transistors is critical to enable very large-scale integration (VLSI) of monolithic 3-D (M3D) integrated circuit (IC) in complementary metal oxide semiconductor (CMOS) backend of line (BEOL). Atomic layer deposition (ALD) deposited ZnO is an attractive candidate due to its excellent electrical properties, ability to be low-temperature processed well below copper interconnect thermal budget of 400°C, and conformal sidewall deposition for novel 3D architecture. However, challenges in demonstrating stable and high-performance logic circuits derived from ALD ZnO process still exist. To address this, we present an optimized ALD deposited ZnO thin-film transistor achieving a record field-effect and intrinsic mobility (μ_{FE}/μ_0) of 85/140 $\text{cm}^2/\text{V}\cdot\text{s}$, and exhibiting excellent time stability. These are all accompanied by a high current on-off ratio $> 10^8$, D_{it} of $2.45 \times 10^{11} \text{ eV}^{-1}\text{cm}^{-2}$, hysteresis $< 52 \text{ mV}$, and low positive threshold voltage suitable for ultra-low power circuit design. The ZnO TFT was also integrated with HfO_2 RRAM in a 1 kbit (32×32) 1T1R array, demonstrating functionalities and decent repeatability for RRAM switching. Furthermore, in order to co-design for future technology requiring high performance BEOL circuitries implementation, we first developed a spice-compatible model capturing the electrical characteristics of the ZnO TFTs. We then simulated and experimentally shown the designs of various ZnO TFT-based inverters, and 5-stage ring oscillators that can achieve a working frequency exceeding 10’s of MHz. Our results thus provide a valuable framework for the future design of high-performance ZnO-based circuitries beyond conventional CMOS technology.”

2. Introduction: The purpose and issue should be clearly described.

We have revised the introduction to better address the inadequacies.

Change in manuscript: **Yes**

Location of Change:

Page 2 (**Introduction**)

“Among them, oxide semiconductors, such as indium gallium zinc oxide (IGZO)¹³⁻¹⁵, indium oxide (In_2O_3)¹⁶, and zinc oxide (ZnO)¹⁷⁻²², are well poised to be competitive n-channel materials beyond silicon due to their low thermal budget process, good transparency, process maturity for large scale deposition, and decent electrical properties, such as high carrier mobility, wide bandgap, low gate leakage^{23,24}. These merits make them suitable for backend-of-line (BEOL) integration as memory drivers

in memory centric computing cells or high performance thin-film transistor (TFT)-based BEOL logic circuitries (**Fig. 1a**)^{7,25}. With the increasing interest to realize new computing architecture with new functionalities and enhanced computing power, the development of high-performance oxide-based transistors to enable very large-scale integration (VLSI) of monolithic 3-D (M3D) integrated circuit (IC) in CMOS BEOL is timely.”

, and

“While there have been reports on high performance ZnO TFT²⁷, the reproducibility of results is still challenging due to the difficult process control as well as ensuring long term stability of the ZnO layer due to its hygroscopic nature. Further, among the popular deposition techniques, ALD approach is attractive for its low temperature process (typically < 300°C), accurate stoichiometry, thickness and uniformity control, and conformal sidewall deposition for M3D integration of vertically integrated architectures^{28,29}. However, despite the promises, reports on high performance ALD ZnO TFT, and a systematic guidance to implement it in BEOL compatible logic circuit are still lacking, calling the need to further address these aspects.”

L85-90: References need to be inserted.

Thank you for pointing this out. We have added the following references.

- Shao, L. L. *et al.* Compact Modeling of Carbon Nanotube Thin Film Transistors for Flexible Circuit Design. *Des Aut Test Europe*, 491-496 (2018).
- Huang, T.-C. *et al.* Pseudo-CMOS: A Design Style for Low-Cost and Robust Flexible Electronics. *IEEE Transactions on Electron Devices* **58**, 141-150 (2011). <https://doi.org:10.1109/ted.2010.2088127>.

Fig. 1: TFT Channel. In Fig. 1c)-1), the gate contact is not shown, which might appear to be unusual. Detailed condition of ZnO deposition with ALD might be helpful for readers to understand the fabrication process.

Fig. 2a shows the zoomed-in SEM image of the ZnO TFT channel, with the gate contact. The overlap between gate and source/drain is 1 μm .

We have also included detailed ALD conditions of the ZnO and HfO₂ thin-film deposition to ensure readers can understand the fabrication process better. The details are provided in the methods section.

Change in manuscript: **Yes**

Location of Change:

Page 14 (**Methods – Device fabrication**)

“In HfO₂ ALD process, the pulse time and purge time for TEMA are 1.6s and 10s, while for H₂O are 0.1s and 10s, respectively. The carrier gas flows for TEMA and H₂O are 80 sccm and 100 sccm, respectively. In ZnO ALD process, the pulse time and purge time for DEZ are 0.1s and 4s, while for H₂O are 0.2s and 4s, respectively. The carrier gas flows for DEZ and H₂O are 150 sccm and 200 sccm, respectively. The TEMA source bottle is heated to 120° C, while DEZ source is keeping at room

temperature during deposition process.”

Could you please provide information about the substrate material?

The ZnO TFT devices were fabricated on 285 nm SiO₂ on Si substrate. The details are provided in the manuscript in page 14. (**Methods – Device fabrication**)

How was the ZnO etched in the fabrication process?

The ZnO film was etched by first using photo lithography to define and protect the channel region, followed by buffered oxide etch (BOE) of 8 s to etch the unwanted regions of ZnO thin-film. A careful time etch was employed to ensure that the SiO₂ layer was not damaged during the etching process. The etching time was calibrated separately by using different etching time and checking the etched thickness using ellipsometry. The SiO₂ layer thickness was also measured after BOE etching.

ZnO is highly soluble in alkali and acid solutions, making it important to provide the readers with information about the material variability.

In our fabrication, the photolithography uses AZ5214 photoresist and RZX3038 developer. AZ5214 does not attack the ZnO film and protects the ZnO channel region during the developing process. The developer contains TMAH, 2.38%. The acetone and isopropanol (IPA) are used in cleaning and photoresist removal process, and they both do not attack the ZnO film. We have included this information in the **Methods – Device fabrication** part of the manuscript.

Additionally, defining Ti/Pt on ZnO can be challenging. The authors might have used a strong alkaline developer on the ZnO thin film which damage the semiconductor too much.

As per our response to the earlier comments, the developer is RZX3038, containing 2.38% TMAH. As TMAH can attack ZnO film, during the source/drain photolithography process, we finely and accurately control the developing time to alleviate the damage to ZnO. When the developing time is 38s, the photoresist can be completely removed, while maintaining negligible damage to the ZnO film. We have included this information in the **Methods – Device fabrication** part of the manuscript.

Fig. 2c: XRD diffraction peaks should be labeled. Why do the XRD patterns differ significantly from each other?

We apologized for missing out on the labels. The diffraction peaks corresponding to the different phase and orientations are labeled. We have also supplemented the analysis of XRD patterns between the different films in the revised manuscript.

Change in manuscript: **Yes**

Location of Change:

Page 5 (**Fig. 2c, d**)

Fig. 2 | Material characterizations of ZnO channel layer. ... c, d GI-XRD patterns, and average crystallite sizes calculated from the Scherrer's formula respectively, of the 3 different temperatures deposited ZnO films.

Change in manuscript: **Yes**

Location of Change:

Page 5 (**Results and Discussion – Materials characterization**)

“To analyze the crystallinity and the preferred orientation, grazing incidence X-ray diffraction (GI-XRD) scans were obtained for the ZnO thin- films deposited at the 3 different temperatures, as shown in **Fig. 2c**. The XRD patterns show that all the ZnO films are polycrystalline with a hexagonal wurtzite structure, with the peaks identified as (100), (002), (101), (102), (110), (103), (112) phases^{39,40}. The XRD patterns of all the three films exhibit the enhanced intensities for the peaks corresponding to (002) plane, indicating preferential orientation along the c-axis^{40,41}. The 150°C deposited ZnO film shows the preferable (002) orientation with additional (100) and (101) peaks. However, the (100) and (101) peaks weaken significantly, and the (002) peak dominates as the deposition temperature of ZnO films increases. This can be attributed to the low surface energy of (002) plane, which is the most thermodynamically favorable^{40,42,43}. When the deposition temperature of film is high, Zn and O atoms can obtain sufficient energy to transfer themselves into energetically favorable positions, which results in ZnO thin films transiting to the (002) preferred orientation at high deposition temperature⁴⁴⁻⁴⁶. **Fig. 2d** shows the average crystallite sizes of the 3 different ZnO thin-films, which were calculated using the Scherrer's formula (see **Supplementary Note S1** for details). The largest average crystallite size of 5.14 nm is obtained of the 200°C deposited ZnO film, which is favorable for high mobility due to reduced grain boundary scattering⁴⁷.”

d: The calculation of crystallite size should be described in detail with references. There are likely several assumptions made to calculate the crystallite size.

The details of the crystallite size calculations are provided in the **Supplementary Note S1** as follows.

The average crystallite sizes of polycrystalline ZnO thin films under different process temperatures are calculated through Scherrer Equation, shown as,

$$D = \frac{K\gamma}{B\cos\theta}$$

where D is the mean size of the ordered crystalline domains, K is a dimensionless shape factor with a value close to unity ($K = 0.89$), γ is the X-ray wavelength (1.54056 Å), B is the line broadening at half the maximum intensity (FWHM) after subtracting the instrumental line broadening, in radians, θ is the Bragg angle.

The average crystallite sizes of the 3 different temperatures deposited ZnO films are 4.58 nm, 5.14 nm and 4.85nm, respectively. The separate B and θ value of each XRD spectra peak are shown in following table.

150°C deposited ZnO film		200°C deposited ZnO film		250°C deposited ZnO film	
2θ (°)	FWHM (°)	2θ (°)	FWHM (°)	2θ (°)	FWHM (°)
32.04	1.44	32.12	1.83	34.49	1.33
34.45	1.37	34.49	1.39	36.31	1.41
36.24	1.88	36.29	1.66	47.54	1.81
47.61	2.16	47.55	2.09	52.43	1.70
52.40	1.69	52.67	1.86	54.29	1.83
54.45	2.13	54.71	0.77	55.77	2.93
56.45	2.40	55.58	3.32	62.88	2.12
62.89	2.15	62.85	2.26	68.13	2.08
68.23	2.47	68.09	2.46		

e: Reference should be added in the caption for XPS results.

Thank you for pointing this out. We have added in the references in the caption for the XPS results as well for consistency

Added references:

- Lan, J. et al. Improved Performance of HfxZnyO-Based RRAM and its Switching Characteristics down to 4 K Temperature. *Advanced Electronic Materials* 9 (2023). <https://doi.org:10.1002/aelm.202201250>
- Chen, X. et al. Transparent and Flexible Thin-Film Transistors with High Performance Prepared at Ultralow Temperatures by Atomic Layer Deposition. *Advanced Electronic Materials* 5 (2019). <https://doi.org:10.1002/aelm.201800583>
- Hsieh, P. T., Chen, Y. C., Kao, K. S. & Wang, C. M. Luminescence mechanism of ZnO thin film investigated by XPS measurement. *Applied Physics A* 90, 317-321 (2007). <https://doi.org:10.1007/s00339-007-4275-3>
- Chen, X., Wan, J., Wu, H. & Liu, C. ZnO bilayer thin film transistors using H2O and O3 as oxidants by atomic layer deposition. *Acta Materialia* 185, 204-210

(2020). <https://doi.org/10.1016/j.actamat.2019.11.066>

- Chen, X., Wan, J., Gao, J., Wu, H. & Liu, C. Enhanced Negative Bias Illumination Stability of ZnO Thin Film Transistors by Using a Two-Step Oxidation Method. *IEEE Transactions on Electron Devices* 69, 2404-2408 (2022). <https://doi.org/10.1109/ted.2022.3159284>

Change in manuscript: **Yes**

Location of Change:

Page 5 (**Fig. 2**)

Fig. 2 | Material characterizations of ZnO channel layer. ... e, f O1s XPS spectra of the ZnO channel layers deposited at 3 temperatures as indicated, and plot showing the Zn-O, V_O, and -OH atomic percentages as a function of the 3 different temperatures used^{12,17,49-51}.

Fig. 3a: What is the reason for the difference observed in the ZnO TFT characteristics with different temperature?

The ZnO TFTs fabricated at different temperatures possess different amount of trap states. Trap states are known to detrimentally affect the performance of transistors by reducing the field-effect mobility (μ_{FE}) as well as an undesired over shift in the V_{TH} due to the presence of unstable stray charges that could enter or leave the trap states. This phenomenon has been reported in literature. We have revised the manuscript and include appropriate references to further explain this observation.

Added references:

- Zhang, L., Li, J., Zhang, X. W., Jiang, X. Y. & Zhang, Z. L. High performance ZnO-thin-film transistor with Ta₂O₅ dielectrics fabricated at room temperature. *Applied Physics Letters* 95 (2009). <https://doi.org/10.1063/1.3206917>
- Li, H. *et al.* High-Performance ZnO Thin-Film Transistors Prepared by Atomic Layer Deposition. *IEEE Transactions on Electron Devices* 66, 2965-2970 (2019). <https://doi.org/10.1109/ted.2019.2915625>
- Yang, J. *et al.* Characteristics of ALD-ZnO Thin Film Transistor Using H₂O and H₂O₂ as Oxygen Sources. *Advanced Materials Interfaces* 9 (2022). <https://doi.org/10.1002/admi.202101953>

Also, the observation of the hump in the transfer curve has been reported in several

literature where in unoptimized growth conditions, results in the formation of dual conduction channels in the channel layer, i.e., a main channel and a parasitic channel that turns on at different voltage. We have revised the manuscript to include more explanation and appropriate references for better clarity.

Added references:

- Yang, J. *et al.* Investigation of a Hump Phenomenon in Back-Channel-Etched Amorphous In-Ga-Zn-O Thin-Film Transistors Under Negative Bias Stress. *IEEE Electron Device Letters* **38**, 592-595 (2017). <https://doi.org:10.1109/led.2017.2686898>
- Kim, W.-S. *et al.* Abnormal behavior with hump characteristics in current stressed a-InGaZnO thin film transistors. *Solid State Electron* **137**, 22-28 (2017). <https://doi.org:10.1016/j.sse.2017.08.001>
- Teng, T., Hu, C.-F., Qu, X.-P. & Wang, M. Investigation of the anomalous hump phenomenon in amorphous InGaZnO thin-film transistors. *Solid State Electron* **170** (2020). <https://doi.org:10.1016/j.sse.2020.107814>
- Li, Q. *et al.* Structural Engineering Effects on Hump Characteristics of ZnO/InSnO Heterojunction Thin-Film Transistors. *Nanomaterials (Basel)* **12** (2022). <https://doi.org:10.3390/nano12071167>

Change in manuscript: **Yes**

Location of Change:

Page 7 (**Results and Discussion – ZnO TFT electrical characterization**)

“The 150 °C deposited ZnO TFT exhibits the lowest drive current, lowest μ_{FE} (3.63 cm²/V·s), and largest V_{TH} (2.5 V) as compared to the TFTs fabricated at higher temperatures. This suggests the detrimental effect of higher trap states, which allows unstable stray charges entering or leaving the traps, resulting in reduced μ_{FE} and overly positive shift in the V_{TH} ^{40,45,51}.”

Location of Change:

Page 7 (**Results and Discussion – ZnO TFT electrical characterization**)

“On the other hand, the ZnO TFTs deposited at 220°C shows the emergence of a small hump in the subthreshold region of the transfer curve (**Supplementary Figure S4**), while a distinct hump phenomenon appears in the 250 °C deposited ZnO TFT that is accompanied by a negative shift of V_{TH} , observable in all devices fabricated at this temperature. This phenomenon has been examined in several works, and can be attributed to the existence of dual conduction channels, i.e., a main channel and a parasitic channel that turn on a different voltage as a result of unoptimized growth conditions⁵²⁻⁵⁵.”

Could you provide information about the thickness employed for the TFT characteristics in Fig. 3a?

In all 3 different ZnO thin-films deposited, they have a nominal thickness of 15 nm as characterized by ellipsometer with good fit. The thin-film deposited at 200°C is cross-verified with TEM results presented in Fig. 2b, confirming the accuracy of our ellipsometer measurements. This information is presented in page 4 of the manuscript

(Results and Discussion - Fabrication of TFT and circuits).

What are the parameters used to calculate μ_{FE} , such as relative permittivity?

The mobility extraction method is provided in the **Supplementary Note S2** as follows.

The field-effect mobility (μ_{FE}) and intrinsic mobility (μ_o) are calculated using Eq (1) and (2),

$$\mu_{FE} = \frac{G_m}{C_{ox} \frac{W}{L} V_{DS}} \quad (1)$$

$$\mu_o = \frac{\mu_{FE}}{1 - \frac{2R_{contact}}{R_{total}}} \quad (2)$$

where G_m is the transconductance extracted from transfer curves of TFT, W and L are the channel width (10 μm) and length (5 μm), V_{DS} is the applied drain voltage, C_{ox} is calculated from an extracted dielectric constant of ~ 15 using a metal-insulator-metal (MIM) capacitor, and R_{total} and $R_{contact}$ are acquired from Fig. 3b in main text.

c: Description about the inserted graph should be added in the caption.

We apologized for missing out on the description for the inset, we have revised the caption accordingly.

Change in manuscript: **Yes**

Location of Change:

Page 7 (**Fig. 3**)

Fig. 3 | ZnO TFT electrical performance. ... c I_D - V_{GS} family of curves (dual sweep) measured over different V_{DS} (0.2 to 2V). A high current on-off ratio up to $\sim 10^8$ is obtained with SS of 110 mV/dec and hysteresis < 52 mV. d Output (I_D - V_{DS}) family of curves (dual sweep) measured over different V_{GS} (0 to 3.5 V) of the same TFT. The zoomed-in region of the plots in 3c and 3d (circled) are shown in the inset respectively, indicating the small hysteresis measured of our fabricated TFT.

d: Description about the inserted graph should be added in the caption.

We apologized for missing out on the description for the inset, we have revised the caption accordingly.

Fig. 7: Are the mobility values consistent between the ring oscillator and TFT characteristics?

Yes. The TFTs, inverters and oscillators are fabricated in the same samples, with the consistent process steps. Also from simulation, the same mobility values are input, correlated to experimental results with excellent agreement. This information is presented in page 4 of the manuscript (**Results and Discussion - Fabrication of TFT and circuits**).

“In the fabrication, the process flow was basically the same as the ZnO TFT, with only the additional step of via hole opening before top electrode deposition for overlapping bottom and top interconnects.”

The text should include a description of D_{it} . Consider discussing oxide TFTs such as ZnO TFT, IGZO, and BEOL-IGZO, etc. What is the reason for the enhanced mode TFT and high μ_{FE} ?

We have revised the manuscript to include more discussion on the D_{it} , enhanced mode TFT and the high mobility achieved in our ZnO TFT as follows:

As presented in Eq. 6 in our supplementary Table S1

$$SS \approx \frac{kT}{q} \ln 10 \cdot (1 + qD_{it}/C_{ox}) \text{ --- (Eq. 6)}$$

, the SS of the TFT is proportionate to the temperature (T) and D_{it} , and inversely proportionate to the oxide capacitance (C_{ox}). In our measurements, the D_{it} can be directly inferred from the SS extracted from the transfer curves of the ZnO TFTs while possessing knowledge on the C_{ox} . The interface property between the ZnO channel and HfO_2 dielectric is crucial for the transistor operation. It has been well-established that the existence of oxygen vacancy (V_O) at the interface should be treated as the origin of interface states. Charge transfer will occur between the channel carriers and interface states, which leads to degraded transport and poorer electrostatic control over the channel. Therefore, low density of states of interface traps (D_{it}) is highly desired to enable a transistor with enhancement-mode, hysteresis-free, high mobility and suppressed SS characteristics. We have included more discussion in the revised manuscript for better clarity.

As in the case of IGZO oxide semiconductor, the depletion-mode FET with significant charge-carrier density in the channel at $V_{GS} = 0$ V is undesirable for applications, since a negative gate-to-source voltage is required to turn off the transistor. However, different from IGZO which is amorphous in nature, the ZnO film studied in this work is polycrystalline, supported by XRD characterization in **Fig. 2c, d**. This results in improved mobility due to suppressed carrier scattering, indicating the potential of ZnO. In addition, our well-controlled ALD deposition process of ZnO film eliminates the interface dipoles and trap states, which reduces the mobile charges present in the channel significantly. A nearly charge-neutral ZnO- HfO_2 interface can thus be obtained to enable the desirable enhancement-mode characteristics.

Change in manuscript: **Yes**

Location of Change:

Page 10 - 11 (**Modelling and Simulation**)

“Furthermore, the expression for the subthreshold swing (SS) can be reduced as **Eq. 6** (Supplementary Table S1), from which the density of states for the interface traps (D_{it}) can be obtained. From Eq. 6 in Supplementary Table S1, the SS of the TFT is

proportionate to the temperature (T) and D_{it} , and inversely proportionate to the oxide capacitance (C_{ox}). In our measurements, the D_{it} can be directly inferred from the SS extracted from the transfer curves of the ZnO TFTs while possessing knowledge on the C_{ox} . The interface property between the ZnO channel and HfO₂ dielectric is crucial for the transistor operation. It has been well-established that the existence of V_O at the interface should be treated as the origin of interface states^{65,66}. Charge transfer will occur between the channel carriers and interface states, which leads to degraded transport and poorer electrostatic control over the channel. Therefore, low density of states of interface traps (D_{it}) is highly desired to enable a transistor with enhancement-mode, hysteresis-free, high mobility and suppressed SS characteristics.”

, and

“As in the case of IGZO oxide semiconductor, the depletion-mode FET with significant charge-carrier density in the channel at $V_{GS} = 0$ V is undesirable for applications, since a negative gate-to-source voltage is required to turn off the transistor. However, different from IGZO which is amorphous in nature, the ZnO film studied in this work is polycrystalline, supported by XRD characterization in **Fig. 2c, d**. This results in improved mobility due to suppressed carrier scattering, indicating the potential of ZnO. In addition, our well-controlled ALD deposition process of ZnO film eliminates the interface dipoles and trap states, which reduces the mobile charges present in the channel significantly. A nearly charge-neutral ZnO-HfO₂ interface can thus be obtained to enable the desirable enhancement-mode characteristics.”

REVIEWERS' COMMENTS

Reviewer #1 (Remarks to the Author):

I reviewed the revised version of the manuscript titled "CMOS Backend-of-Line Compatible Memory Array and Logic Circuitries Enabled by High-Performance Atomic Layer Deposited ZnO Thin-Film Transistor" and can confirm that the authors responded diligently to all questions, accepted suggestions from the reviewers, and made outstanding improvements. I recommend this manuscript for publication because it offers a comprehensive analysis of ZnO thin-film transistors and circuits based on these transistors. The study outlined in this manuscript represents a significant advancement in the state of the art for ZnO TFTs.

Reviewer #2 (Remarks to the Author):

The authors responded promptly, and the manuscript has been revised based on the reviewers' comments. However, there is some ambiguity in the discussion regarding crystallite size.

The calculation of crystallite size is conducted based on diffraction peaks in all crystallographic directions. This approach may assume that the crystallite size is the same in all directions. Specifically, the ZnO film is oriented in the <002> direction vertically to the substrate surface. The average crystallite size must have a distribution according to the crystal's direction.

Therefore, it is advisable to explicitly state any assumptions made in the text.

Author's Response to Reviewer's comments

we sincerely thank the reviewers for the insightful and constructive comments to improve the manuscript, and their positive response to our work. We have revised our manuscript to address the questions and concerns raised by the reviewers. Please find the point-by-point response to the reviewer's comment below in **Blue**, and the revised text in manuscript in **Red**. The changes made in the manuscript are highlighted in **yellow** for reference.

REVIEWER COMMENTS

Reviewer #1 (Remarks to the Author):

I reviewed the revised version of the manuscript titled "CMOS Backend-of-Line Compatible Memory Array and Logic Circuitries Enabled by High-Performance Atomic Layer Deposited ZnO Thin-Film Transistor" and can confirm that the authors responded diligently to all questions, accepted suggestions from the reviewers, and made outstanding improvements. I recommend this manuscript for publication because it offers a comprehensive analysis of ZnO thin-film transistors and circuits based on these transistors. The study outlined in this manuscript represents a significant advancement in the state of the art for ZnO TFTs.

We sincerely thank the reviewer for his positive comments to our work.

Reviewer #2 (Remarks to the Author):

The authors responded promptly, and the manuscript has been revised based on the reviewers' comments. However, there is some ambiguity in the discussion regarding crystallite size.

The calculation of crystallite size is conducted based on diffraction peaks in all crystallographic directions. This approach may assume that the crystallite size is the same in all directions. Specifically, the ZnO film is oriented in the <002> direction vertically to the substrate surface. The average crystallite size must have a distribution according to the crystal's direction.

Therefore, it is advisable to explicitly state any assumptions made in the text.

We sincerely thank the reviewer for his valuable comment. Indeed, in our analysis, the crystallite size calculation was conducted based on all the different diffraction peaks presented in Fig. 2c. This was done by assuming that the average crystallite size is the same in all directions. We have revised our manuscript to explicitly state the assumptions made for the crystallite size calculation in this work, in order to provide clarity to the readers.

Change in manuscript: **Yes**

Location of Change:

Page 6 (**Results and Discussion – Materials characterization**)

"Fig. 2d shows the average crystallite sizes of the 3 different ZnO thin-films, which

were calculated using the Scherrer's formula. The crystallite size calculations are conducted based on the different diffraction peaks measured, and by assuming that the crystallite size is the same in all directions (see Supplementary Note S1 for details)."